# Flexible MOF Generation
# with Torsion-Aware Flow Matching

**Nayoung Kim**    **Seongsu Kim**    **Sungsoo Ahn**
Korea Advanced Institute of Science and Technology (KAIST)
{nayoungkim, seongsu.kim, sungsoo.ahn}@kaist.ac.kr

## Abstract

Designing metal-organic frameworks (MOFs) with novel chemistries is a longstanding challenge due to their large combinatorial space and complex 3D arrangements of the building blocks. While recent deep generative models have enabled scalable MOF generation, they assume (1) a fixed set of building blocks and (2) known local 3D coordinates of building blocks. However, this limits their ability to (1) design novel MOFs and (2) generate the structure using novel building blocks. We propose a two-stage MOF generation framework that overcomes these limitations by modeling both chemical and geometric degrees of freedom. First, we train an SMILES-based autoregressive model to generate metal and organic building blocks, paired with a cheminformatics toolkit for 3D structure initialization. Second, we introduce a flow matching model that predicts translations, rotations, and torsional angles to assemble the blocks into valid 3D frameworks. Our experiments demonstrate improved reconstruction accuracy, the generation of valid, novel, and unique MOFs, and the ability to create novel building blocks. Our code is available at https://github.com/nayoung10/MOFFlow-2.

## 1   Introduction

Metal-organic frameworks (MOFs) are highly crystalline materials known for their permanent porosity, structural versatility, and broad applications in fields such as gas storage and separations [1, 2], catalysis [3], and drug delivery [4]. MOFs are designed by assembling chemical building blocks, i.e., metal clusters and organic linkers; their properties (e.g., pore size) can be *tuned* by swapping or modifying these components. This tunability has enabled the synthesis of over 100,000 distinct MOF structures to date [5], yet the theoretical space of possible MOFs is vastly larger, on the order of millions of structures [6]. Exploring this large design space is a grand challenge in materials science.

Computational modeling of MOFs, i.e., 3D structure prediction and generating new MOFs, poses significant challenges because of their structural complexity. Traditional methods, such as ab initio random search [7], attempt to design new MOFs or predict their structures by iteratively proposing combinations of building blocks and evaluating their stability using energy-based criteria. However, MOFs typically contain hundreds of atoms per unit cell, rendering these iterative approaches computationally prohibitive. Even state-of-the-art deep generative models developed for inorganic materials [8, 9] struggle to scale to the size and complexity of MOFs [10].

In response, researchers have proposed MOF-specific generative models such as MOFDiff [11] and MOFFlow [10]. While these methods represent significant progress, their design choices still constrain both chemical diversity and structural fidelity. Specifically, they (1) rely on a fixed set of predefined building blocks and (2) assume these building blocks are rigid. As a result, the design space is limited to recombinations of known components with fixed 3D conformations, hindering the discovery of novel chemistries and overlooking the intrinsic flexibility of organic linkers [12].

39th Conference on Neural Information Processing Systems (NeurIPS 2025).

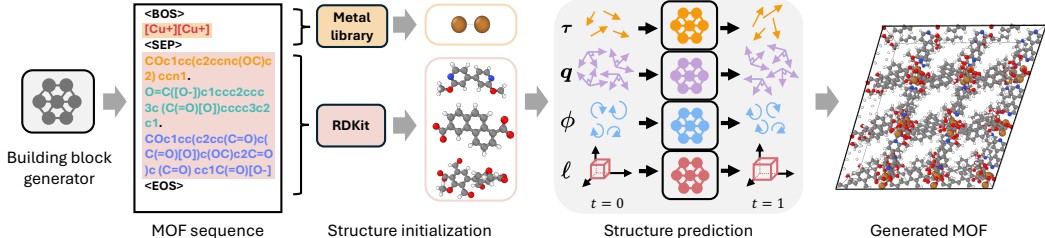

**Figure 1: Overview of MOFFLOW-2.** MOFFLOW-2 is a two-stage generative framework for MOF generation and structure prediction. The first stage uses a building block generator to generate an MOF sequence in SMILES representation, which is initialized to 3D coordinates with the metal library and RDKit. In the second stage, our structure prediction model assembles these building blocks by modeling translation $\tau$, rotation $q$, torsion $\phi$, and lattice $\ell$.

**Contribution.** To address these limitations, we introduce MOFFLOW-2, a two-stage generative framework capable of producing novel MOF chemistries and predicting high-fidelity 3D structures (Figure 1). Unlike prior methods, MOFFLOW-2 does not rely on a fixed building block library or assume rigid conformations.

In the first stage, an autoregressive Transformer generates a canonicalized sequence of MOF building blocks represented as SMILES [13], enabling the creation of entirely novel chemistries. These SMILES can be initialized to 3D structures using cheminformatics tools such as RDKit [14]. In the second stage, a flow-based model predicts the full 3D MOF structure by jointly modeling (1) rotations and translations of building blocks, (2) lattice structure, and (3) torsion angle for rotatable bonds in the organic linkers. Crucially, by explicitly modeling the torsion angle, MOFFLOW-2 captures the conformational flexibility of organic linkers and no longer depends on predefined and rigid 3D conformations. See Table 1 for comparison to prior works.

Table 1: **Comparison of MOF generative models.** *Trans* refers to translation modeling, *Rot* to rotation modeling, *Tor* to torsion angle modeling, and *NBB* to the ability to generate novel building blocks.

| Method | Trans | Rot | Tor | NBB |
|---|---|---|---|---|
| MOFDiff [11] | ✔ | ▲ | ✘ | ✘ |
| MOFFLOW [10] | ✔ | ✔ | ✘ | — |
| **MOFFLOW-2** | ✔ | ✔ | ✔ | ✔ |

We evaluate MOFFLOW-2 on both structure prediction and generative design tasks. For structure prediction, we test the model without providing 3D coordinates of the building blocks – a realistic but difficult setting. MOFFLOW-2 outperforms MOFFlow, demonstrating its ability to model flexible conformations without relying on rigid-body assumptions. For the generation task, MOFFLOW-2 achieves higher validity, novelty, and uniqueness than MOFDiff, and generates MOFs with a more diverse range of properties. Importantly, it can generate MOFs with building blocks not seen during training, going beyond a fixed building block library. These results show that MOFFLOW-2 is effective for both accurate structure prediction and discovering novel MOFs, providing a step towards more automated and flexible MOF design.

## 2    Related work

**Generative models for MOFs.** Unlike inorganic crystals, MOFs present unique challenges due to their large size, often containing hundreds of atoms per unit cell. MOFDiff [11] addresses this by generating coarse-grained structures via diffusion and decoding them using a fixed set of building blocks extracted from training data. Similarly, MOFFlow [10] assumes access to the 3D coordinates of building blocks and treats them as rigid bodies, learning block-level rotations and translations instead of predicting atom-level coordinates. In this work, we propose MOFFLOW-2, a unified and flexible two-stage framework for both MOF generation and structure prediction. Unlike previous methods, MOFFLOW-2 does not rely on a fixed building block library or require 3D coordinates as input, enabling greater chemical diversity and generality.

**Equivariance and generative models.** While SE(3) and E(3)-equivariant generative models have shown promising results on molecular tasks [15–18], recent studies question whether equivariance is essential [19]. Notably, AlphaFold3 [20], Boltz-1 [21], and Proteína [22] achieve state-of-the-art results in modeling biomolecular complexes without using equivariant architectures. Similarly, Orb [23] and ADiT [24] employ non-equivariant neural networks to model interatomic potentials and

generate molecule and crystal structures effectively. Following this trend, MOFFLOW-2 adopts a non-equivariant Transformer architecture.

## 3 Preliminaries

### 3.1 Structural representation of MOFs

**Crystal representation.** A 3D crystal structure is defined by the periodic repetition of the unit cell. A unit cell containing $N$ atoms is represented by the tuple $\mathcal{S} = (\boldsymbol{A}, \boldsymbol{X}, \boldsymbol{\ell})$, where $\boldsymbol{A} = \{a_i\}_{i=1}^N \in \mathcal{A}^N$ is the atom types, with $\mathcal{A}$ being the set of chemical elements; $\boldsymbol{X} = \{x_i\}_{i=1}^N \in \mathbb{R}^{N \times 3}$ is the atomic coordinates; and $\boldsymbol{\ell} = (a, b, c, \alpha, \beta, \gamma) \in \mathbb{R}_+^3 \times [0°, 180°]^3$ is the lattice parameters, where $(a, b, c)$ are the edge lengths and $(\alpha, \beta, \gamma)$ are the angles between them. The lattice parameters $\boldsymbol{\ell}$ can be converted to the standard lattice matrix $L = (l_1, l_2, l_3) \in \mathbb{R}^{3 \times 3}$ where $l_i$ is a lattice vector for each edge. Translating the unit cell along the lattice vectors defines an infinite crystal structure as $\{(a_n, x_n + Lk)|n \in [N], k \in \mathbb{Z}^3\}$ where $k = (k_1, k_2, k_3)^\top \in \mathbb{Z}^3$ is the periodic translation.

**MOF representation.** The modular structure of MOFs allows a block-level representation based on their building blocks. For an MOF with $N$ atoms, we write $\mathcal{S} = (\mathcal{B}_{\text{3D}}, \boldsymbol{q}, \boldsymbol{\tau}, \boldsymbol{\phi}, \boldsymbol{\ell})$, where $\mathcal{B}_{\text{3D}} = \{\mathcal{C}^{(m)}\}_{m=1}^M$ denotes a set of $M$ building blocks. Each building block $\mathcal{C}^{(m)} = (A^{(m)}, Y^{(m)})$ contains $N_m$ atoms with atom types $A^{(m)} \in \mathcal{A}^{N_m}$ and local coordinates $Y^{(m)} \in \mathbb{R}^{N_m \times 3}$. We represent the 3D structural degrees of freedom with rotations $\boldsymbol{q} = \{q^{(m)} \in SO(3)\}_{m=1}^M$, translations $\boldsymbol{\tau} = \{\tau^{(m)} \in \mathbb{R}^3\}_{m=1}^M$, and torsions $\boldsymbol{\phi} = \{\phi^{(m)} \in SO(2)^{P_m}\}_{m=1}^M$, where $P_m$ is the number of rotatable bonds in block $\mathcal{C}^{(m)}$. Following Jing et al. [16], we define a bond as rotatable if severing the bond creates two substructures, each with at least two atoms. The global coordinates $\boldsymbol{X}$ can be reconstructed by applying these transformations to local building block coordinates as $X^{(m)} = (q^{(m)}, \tau^{(m)}, \phi^{(m)}) \cdot Y^{(m)}$ and concatenating them $\boldsymbol{X} = \text{Concat}(X^{(1)}, \ldots, X^{(M)})$.

We also consider 2D molecular representations of MOF building blocks, denoted as $\mathcal{B}_{\text{2D}} = \{\mathsf{G}^{(m)}\}_{m=1}^M$. Each graph $\mathsf{G}^{(m)} = (\mathsf{V}^{(m)}, \mathsf{E}^{(m)})$ consists of nodes representing atoms and edges corresponding to bonds in organic linkers. These bonds are inferred using OpenBabel [25] and RDKit [14] by analyzing the 3D structure $\mathcal{C}^{(m)}$ of the building block.

### 3.2 Flow matching on Riemannian manifolds

Flow matching (FM) trains a CNF by learning a vector field that transports samples from an arbitrary prior distribution $p_0$ to the target distribution $q$. Since MOFFLOW-2 operates on $SO(2)$ and $SO(3)$, we here introduce flow matching generalized to Riemannian manifolds [26].

**Flows on Riemannian manifolds.** Here, we define CNF on Riemannian manifolds. Consider a smooth connected Riemannian manifold $\mathcal{M}$ with metric $g$, where each point $x \in \mathcal{M}$ has an associated tangent space $\mathcal{T}_x \mathcal{M}$ with inner product $\langle \cdot, \cdot \rangle_g$. A flow $\phi_t(x) : \mathcal{M} \to \mathcal{M}$ is defined by the ordinary differential equation (ODE) $\frac{d}{dt}\phi_t(x) = u_t(\phi_t(x)), \phi_0(x) = x \sim p_0$ where $t \in [0, 1]$ and $u_t(x) \in \mathcal{T}_x \mathcal{M}$ is the corresponding time-dependent vector field. We say that $u_t$ generates probability paths $p_t(x)$ if $p_t(x) = [\phi_t]_\# p_0(x)$, where $\#$ denotes the pushforward operator.

**Riemannian flow matching.** The goal of flow matching is to learn a time-dependent vector field $v_t(x; \theta)$ that transports an arbitrary initial distribution $p_0(x)$ to $p_1(x)$ that closely approximates the target distribution $q(x)$. Given a reference vector field $u_t(x)$ that transports $p_0$ to $q$, we can train the vector field $v_t(x; \theta)$ by regressing it directly towards $u_t(x)$ with the flow matching objective:

$$\mathcal{L}_{\text{FM}}(\theta) = \mathbb{E}_{t \sim \mathcal{U}(0,1), x \sim p_t(x)} \left[ \|v_t(x; \theta) - u_t(x)\|_g^2 \right], \tag{1}$$

where $\mathcal{U}(0, 1)$ is the uniform distribution on $[0, 1]$, $p_t(x)$ the probability distribution at time $t$, and $\| \cdot \|_g$ is the norm induced by the Riemannian metric $g$.

While the vector field $u_t(x)$ is intractable, flow matching introduces a conditional formulation that yields a tractable and equivalent training objective [27]. Specifically, the probability paths $p_t(x)$ and vector field $u_t(x)$ can be expressed as marginalization over the conditional probability paths $p_t(x|x_1)$ and conditional vector field $u_t(x|x_1)$:

$$p_t(x) = \int_{\mathcal{M}} p_t(x|x_1) q(x_1) \, d\text{vol}(x_1), \quad u_t(x) = \int_{\mathcal{M}} u_t(x|x_1) \frac{p_t(x|x_1) q(x_1)}{p_t(x)} \, d\text{vol}(x_1), \tag{2}$$

where $p_t(x|x_1)$ has boundary conditions $p_0(x|x_1) = p_0(x)$ and $p_1(x|x_1) \approx \delta(x - x_1)$ and $u_t(x|x_1)$ generates $p_t(x|x_1)$. Then we can train $v_t(x; \theta)$ with the conditional flow matching (CFM) objective:

$$\mathcal{L}_{\text{CFM}}(\theta) = \mathbb{E}_{t \sim \mathcal{U}(0,1), x_1 \sim p_1(x), x \sim p_t(x|x_1)} \left[ \|v_t(x; \theta) - u_t(x|x_1)\|_g^2 \right]. \tag{3}$$

# 4 Autoregressive building block generation

In this section, we introduce our model for generating the MOF building blocks, which are used by the structure prediction model to generate the final 3D structure. Formally, we model the joint distribution as $p_\theta(q, \tau, \ell, \phi, \mathcal{B}_{\text{3D}}) = p_\theta(\mathcal{B}_{\text{3D}})p_\theta(q, \tau, \ell, \phi | \mathcal{B}_{\text{3D}})$ where $\mathcal{B}_{\text{3D}}$ denotes the 3D building block representations. To this end, we train a SMILES generative model $p_\theta(\mathcal{B}_{\text{2D}})$ to generate the 2D graph structure of each building block. Then we use pre-built metal building block library and RDKit [14] to obtain $\mathcal{B}_{\text{3D}}$, the initial 3D coordinates of the building blocks.

## 4.1 Autoregressive SMILES generation

The building block generator $p_\theta(\mathcal{B}_{\text{2D}})$ is an autoregressive model that generates MOF sequences, which contain metal clusters and organic linkers represented as SMILES [13]. This section outlines the definition of an MOF sequence, the training objective, and the implementation details.

**Defining MOF sequence.** Given an MOF with $M_1$ metal clusters $m_1, \ldots, m_{M_1}$ and $M_2$ organic linkers $o_1, \ldots, o_{M_2}$, we impose a canonical ordering on the building blocks to ensure a consistent sequence representation. Specifically, (1) Metal building blocks precede organic ones, separated by a special token `<SEP>`, (2) Within each group (metal or organic), building blocks are sorted by ascending molecular weight and separated with the symbol `"."`, and (3) The full sequence is wrapped with start and end tokens `<BOS>` and `<EOS>`. The resulting sequence is $\mathcal{B}_{\text{2D}} = \left[ \text{<BOS> m\_1.m\_2.} \cdots \text{.m\_M1 <SEP> o\_1.o\_2.} \cdots \text{.o\_M2 <EOS>} \right]$, where `m_i` and `o_j` denote the SMILES string of metal and organic building blocks, respectively.

Next, Given the canonical MOF sequence $\mathcal{B}_{\text{2D}}$, we tokenize it into a sequence of discrete symbols $[b_1, \ldots, b_S]$ by mapping substrings (e.g., atoms, bonds, or special tokens) into elements of a fixed vocabulary. We use a SMILES-aware regular expression [28] for tokenization, resulting in a vocabulary of size 59. See Appendix A.1 for an example of the tokenization process.

**Training and model architecture.** We model the sequence using an autoregressive language model $p_\theta(b_1, \ldots, b_S) = \prod_{s=1}^S p_\theta(b_s | b_{<s})$ with each token $b_s$ conditioned on the preceding tokens $b_{<s}$. We train the model with the standard maximum log-likelihood objective.

## 4.2 3D coordinate initialization

Here, we describe our algorithm for initializing the building block 3D coordinates $\mathcal{B}_{\text{3D}}$ from the 2D SMILES representation $\mathcal{B}_{\text{2D}}$. These 3D coordinates serve as input to the structure prediction model. We use different initialization strategies for metal clusters and organic linkers due to their distinct structural characteristics.

Due to their complex chemistry, only a limited set of metal clusters is commonly used in MOFs [12]. Based on this observation, we construct a metal library of averaged 3D coordinates from the training data. Specifically, we group metal clusters by their canonical SMILES, select a random reference structure within each group, align all other structures to the reference using root mean square distance (RMSD) minimization, and compute the average of the aligned coordinates.[1]

In contrast, organic linkers exhibit greater structural diversity and flexibility [12], making the template-based approach impractical. Therefore, we use a cheminformatics toolkit (i.e., RDKit [14]) to initialize standard bond lengths and angles by optimizing the MMFF94 force field [29]. However, the resulting structures are only approximate, as the torsion angles around the rotatable bonds remain highly variable [16]. We therefore treat these torsion angles as learnable variables and model them with our structure prediction module to capture the full conformational flexibility of organic linkers.

---

[1]Appendix B confirms this – our dataset has only 7–8 metal types with low RMSD variability.

# 5 MOF structure prediction with torsional degrees of freedom

Our MOF structure prediction model $p_\theta(\boldsymbol{q}, \boldsymbol{\tau}, \boldsymbol{\ell}, \boldsymbol{\phi} | \mathcal{B}_{3D})$ is a flow-based model that predicts the 3D structural degrees of freedom – rotations $\boldsymbol{q}$, translations $\boldsymbol{\tau}$, torsion angles $\boldsymbol{\phi}$, and lattice parameters $\boldsymbol{\ell}$ – conditioned on the initialized 3D building block structure $\mathcal{B}_{3D}$. In this section, we present the training algorithm based on Riemannian flow matching, a Transformer-based architecture for predicting each structural component, and a canonicalization procedure for rotations and torsions to ensure consistent targets during training.

Importantly, similar to MOFFLOW [10], our structure prediction model MOFFLOW-2 can be used independently of the building block generator. In fact, MOFFLOW-2 is more general, as it can predict structures directly from the 2D building block $\mathcal{B}_{2D}$, without requiring known 3D conformations. This is more practical since the 3D structures are often unavailable or difficult to obtain in the real world.

## 5.1 Training algorithm

**Prior distribution.** We define the priors over each structural component as follows. Rotations and torsions are sampled independently from uniform distributions on their respective manifolds: $q \sim \mathcal{U}(SO(3))$ and $\phi \sim \mathcal{U}(SO(2))$. Translations are drawn from a standard normal distribution with the center of mass removed: $\boldsymbol{\tau} \sim \mathcal{N}(0, I_M)$. For the lattice parameters, the lengths $(a, b, c)$ are sampled independently from log-normal distributions: $(a, b, c) \sim \mathrm{LogNormal}(\boldsymbol{\mu}, \boldsymbol{\sigma})$, where $\boldsymbol{\mu} = (\mu_a, \mu_b, \mu_c)$ and $\boldsymbol{\sigma} = (\sigma_a, \sigma_b, \sigma_c)$ are estimated by maximum likelihood objective; the lattice angles $(\alpha, \beta, \gamma)$ are sampled uniformly from the range $[60°, 120°]$ [9, 10].

**Conditional flows, vector fields, and flow matching loss.** Following Chen and Lipman [26], we define conditional flows on each manifold as geodesic interpolations between a prior sample $\boldsymbol{z}_0$ and a target data $\boldsymbol{z}_1$, where $\boldsymbol{z}_1$ represents one of translation, rotation, torsion, or lattice parameter. Specifically, we define the flow as $\boldsymbol{z}_t = \exp_{\boldsymbol{z}_0}(t \log_{\boldsymbol{z}_0}(\boldsymbol{z}_1))$, where $t \in [0, 1]$ and $\log(\cdot)$, $\exp(\cdot)$ denote logarithm map and exponential map on respective manifolds (see Appendix C.1 manifold-specific details). This allows parameterizing the vector field by $v_t(\boldsymbol{z}_t; \theta) = \log_{\boldsymbol{z}_t}(\hat{\boldsymbol{z}}_1)/(1 - t)$ where the neural network predicts the clean data $\hat{\boldsymbol{z}}_1$ from intermediate state $\boldsymbol{z}_t$. Furthermore, the conditional vector field is given by $u_t(\boldsymbol{z}_t | \boldsymbol{z}_1) = \log_{\boldsymbol{z}_t}(\boldsymbol{z}_1)/(1 - t)$.

Given the closed-form expression of the target conditional vector field, we parameterize the model as

$$(\Delta\hat{\boldsymbol{q}}_{t \to 1}, \hat{\boldsymbol{\tau}}_1, \hat{\boldsymbol{\phi}}_1, \hat{\boldsymbol{\ell}}_1) = \mathcal{F}_\theta(\boldsymbol{A}, \boldsymbol{X}_t, \boldsymbol{\ell}_t), \tag{4}$$

where $\boldsymbol{X}_t$ is the noisy coordinate generated by applying rotation $\Delta\boldsymbol{q}_{1 \to t}$, translation $\boldsymbol{\tau}_t$, and torsion $\boldsymbol{\phi}_t$ to the clean $\boldsymbol{X}_1$ (see Appendix C.2 for precise definition); $\Delta\boldsymbol{q}_{t \to 1} := \boldsymbol{q}_t \boldsymbol{q}_1^\top$ is the rotation that aligns $\boldsymbol{X}_t$ back to $\boldsymbol{X}_1$; and the hat symbol $\hat{}$ indicates the model's prediction of each variable. Under this formulation, we can rewrite the training objective as:

$$\mathcal{L}(\theta) = \lambda_1 \|\Delta\hat{\boldsymbol{q}}_{t \to 1} - \Delta\boldsymbol{q}_{t \to 1}\|_2^2 + \lambda_2 \|\hat{\boldsymbol{\tau}}_1 - \boldsymbol{\tau}_1\|_2^2 + \lambda_3 \|\hat{\boldsymbol{\phi}}_1 - \boldsymbol{\phi}_1\|_2^2 + \lambda_4 \|\hat{\boldsymbol{\ell}}_1 - \boldsymbol{\ell}_1\|_2^2, \tag{5}$$

where $\lambda_1, \lambda_2, \lambda_3, \lambda_4$ are scaling coefficients for rotation, translation, torsion, and lattice, respectively. We present detailed training and inference algorithms in Appendix C.3.

## 5.2 Model architecture

Our structure prediction model is built on a non-equivariant Transformer backbone. The architecture comprises an initialization module, an interaction module, and an output module (Figure 2)

**Initialization module.** The initial node embeddings $h_i^{(0)} \in \mathbb{R}^D$ for $i$-th atom is computed as $h_i^{(0)} = [E_\theta(a_i), f(a_i), x_i, \boldsymbol{\ell}_t, \varphi(t)] + \varphi(k)$, where $E_\theta(a_i)$ is a learnable atom type embedding, $f(a_i)$ is a binary feature vector encoding atom-level features such as aromaticity [15], and $\varphi(\cdot)$ is sinusoidal embedding function, $t$ is timestep, and $k$ is the index of the building block that contains the $i$-th atom. The building block index $k$ resolves ambiguity between blocks with identical SMILES by lexicographic ordering similar to Section 4.1. Full details are provided in Appendix D.1.

**Interaction module.** We adopt the Transformer encoder architecture for the 3D molecular graph $\mathcal{G} = (\mathcal{V}, \mathcal{E})$, where $\mathcal{V}$ represents atoms and $\mathcal{E}$ consists of edges between atoms within a specified cutoff threshold. For each edge $(i, j) \in \mathcal{E}$, we define the edge feature as $e_{ij} = [b(i, j), \mathrm{RBF}(\|x_i - x_j\|_2)]$,

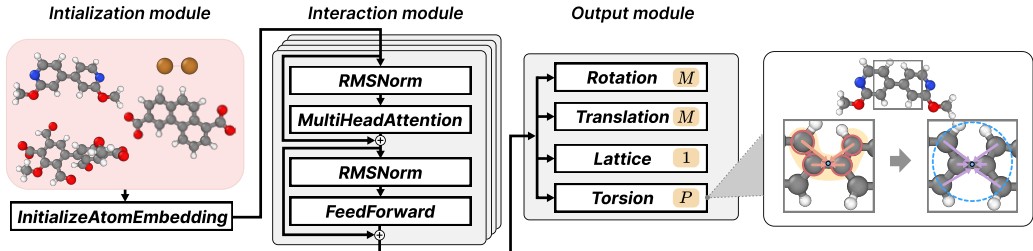

Figure 2: **Structure prediction model architecture.** The model consists of three main modules: an initialization module that encodes atom features; an interaction module based on a Transformer encoder; and an output module with four prediction heads for each structural component – i.e., rotation (dimension $M$, the number of building blocks), translation ($M$), lattice parameters (1), and torsion angles (dimension $P$, the number of rotatable bonds). Notably, torsion angles are predicted by first constructing rotatable bond features with four corresponding dihedral atoms and updating the feature with attention to nearby atoms.

where $b(i, j) \in \{0, 1\}^4$ is a one-hot encoding of the bond type (single, double, triple, aromatic), or all zeros if no bond exists [15], and $\mathrm{RBF}(\cdot)$ is a radial basis function for interatomic distance.

To compute self-attention scores, we first compute the query, key, and value for each target atom $i$ and its neighbor atom $j$ as: $\mathbf{q}_i \leftarrow \mathrm{Linear}_Q(h_i)$, $\mathbf{k}_{ij} \leftarrow \mathrm{Linear}_K([h_j, e_{ij}])$, $\mathbf{v}_{ij} \leftarrow \mathrm{Linear}_V([h_j, e_{ij}])$. Then the attention score from $i$ to $j$ is computed as $a_{ij} = \mathrm{Softmax}_{j \in \mathcal{N}(i)}(\mathbf{q}_i^\top \mathbf{k}_{ij}/\sqrt{D})$, where $D$ is the hidden dimension, and $\mathcal{N}(i)$ is the set of neighbors of atom $i$. Finally, we update the node feature by aggregating the value vectors from its neighbors: $h_i \leftarrow \sum_{j \in \mathcal{N}(i)} a_{ij} \mathbf{v}_{ij}$. We provide additional architectural details and hyperparameters in Appendix D.2.

**Output module.** The output module consists of four heads that predict rotation $\Delta \hat{\boldsymbol{q}}_{1 \to 1}$, translation $\hat{\boldsymbol{\tau}}_1$, torsion $\hat{\boldsymbol{\phi}}_1$, and lattice parameters $\hat{\boldsymbol{\ell}}_1$.

*(Lattice parameter)* To predict the lattice parameter, we mean-pool the node embeddings $\{h_i\}_{i=1}^N$ and apply a two-layer MLP with a GELU activation [30], i.e., set $\hat{\boldsymbol{\ell}} = \mathrm{MLP}(\frac{1}{N} \sum_{i=1}^N h_i)$.

*(Translation)* For the prediction of translation $\boldsymbol{\tau} = \{\tau_m\}_{m=1}^M$, we apply block-wise attention pooling over the nodes in each building block $\mathcal{V}^{(m)} \subseteq \mathcal{V}$, followed by a two-layer MLP, i.e., set $\hat{\tau}_m = \mathrm{MLP}(\mathrm{BlockAttentionPool}(\frac{1}{|\mathcal{V}^{(m)}|} \sum_{i \in \mathcal{V}^{(m)}} h_i))$.

*(Rotation)* Similar to translation, the rotation head applies block-wise attention pooling to node embeddings, followed by a two-layer MLP. Importantly, to represent rotations, we use a continuous 9D matrix representation based on singular value decomposition (SVD)[31–33]. We found this approach to be more stable than common parameterizations – such as Euler angles, axis-angle, or quaternions – which often lead to unstable training due to their discontinuities [31]. Concretely, during training, we supervise the predicted (possibly non-orthogonal) raw matrix output $\Delta \boldsymbol{Q}_{t \to 1}$ using a mean squared loss against the target rotation $\Delta q_{t \to 1}$. During inference, our model projects $\Delta \boldsymbol{Q}_{t \to 1}$ onto the nearest valid rotation matrix by solving the orthogonal Procrustes problem [31]:

$$\Delta \hat{\boldsymbol{q}}_{t \to 1} = \mathrm{Procrustes}(\Delta \boldsymbol{Q}_{t \to 1}) = \underset{R \in SO(3)}{\arg \min} \|R - \Delta \boldsymbol{Q}_{t \to 1}\|_F^2 = U \ \det(1, 1, \det(UV^\top)) \ V^\top, \quad (6)$$

where $U, V$ are from SVD outputs – i.e., $\mathrm{SVD}(\Delta \boldsymbol{Q}_{t \to 1}) = U\Sigma V^\top$.

*(Torsion)* To predict the torsion angles, we first construct invariant features for each rotatable bond and refine them with an attention mechanism. Specifically, for each rotatable bond between atoms $(j, k)$, we first define the dihedral angle using four atoms $(i, j, k, l)$, where atoms $i$ and $l$ are selected according to a consistent canonicalization scheme described in Section 5.3. The corresponding rotatable bond feature $h_{ijkl}$ is constructed by concatenating the atom features and applying a linear transformation in both forward and reverse directions:

$$h_{ijkl} = \mathrm{Linear}_\phi([h_i, h_j, h_k, h_l]) + \mathrm{Linear}_\phi([h_l, h_k, h_j, h_i]), \quad (7)$$

where using both orders ensures that the feature is invariant to the direction of the dihedral. Each rotatable bond feature then attends to neighboring atoms within a 5Å radius using attention, similar to other prediction heads. The result is passed through a two-layer MLP to predict the torsion angle.

### 5.3 Preprocessing with canonicalization and MOF matching

**Canonicalization of rotation.** MOF building blocks often have high symmetry point groups, where multiple rotations produce indistinguishable structures. As a result, there exist multiple "ground-truth" rotations $\Delta q_{t\rightarrow 1}$ that align $X_t$ to $X_1$, leading to unstable training (Figure 3a).

To resolve this ambiguity, we introduce a canonicalization procedure that selects a unique rotation target for each symmetric building block. Given a clean conformation $\mathcal{C}_1 = (A, X_1)$ and a noised conformation $\mathcal{C}_0 = (A, X_0)$, we first identify the set of symmetry-preserving rotations of $\mathcal{C}_0$, defined as $\mathcal{R} := \{\mathfrak{g} \in O(3) | \mathfrak{g} \cdot \mathcal{C}_0 = \mathcal{C}_0\} \cap SO(3)$. We then apply each $\mathfrak{g} \in \mathcal{R}$ to $X_0$ to generate symmetrically equivalent conformations. Among these, we select the one with the lowest RMSD to $X_1$ and recompute the rotation between the two structures with Kabsch alignment [34] (Algorithm 1). This yields the canonical rotation $\Delta q^*_{1\rightarrow 0}$, which we interpolate to obtain the final target rotation $\Delta q_{t\rightarrow 1}$ for any $t \in [0, 1]$.

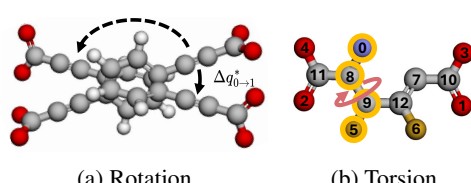

(a) Rotation      (b) Torsion

Figure 3: **Canonicalization.** MOF building blocks often exhibit high symmetry point groups and $\pi$-symmetry bonds, resulting in multiple valid rotations and torsion targets. This ambiguity can lead to unstable training. To resolve this, (a) we assign a unique rotation target by finding the closest rotation in terms of RMSD. (b) For torsions, we uniquely define the neighboring atoms for a rotatable bond with canonical atom rankings of RDKit.

**Canonicalization of torsion.** MOF building blocks frequently contain $\pi$-symmetry bonds associated with distinct rotations that results in an equivalent conformation [17, 35]. Therefore, representing torsion as a relative rotation can lead to multiple valid targets for a single noisy conformation. To address this, we adopt a torsion representation that explicitly specifies the four atoms of the dihedral, with the neighboring atoms selected consistently using RDKit's canonical atom rankings [14]. Concretely, given a rotatable bond between atoms $j$ and $k$, we select atoms $i$ and $l$ as the lowest-ranked neighbors of $j$ and $k$, respectively (Figure 3b).

**MOF matching.** We introduce a preprocessing step to reduce the distributional shift between the training and inference structures. Namely, during training, the model receives DFT-relaxed 3D conformations as input, while at inference, it receives coordinates initialized from the metal building block library and cheminformatics tools (see Section 4.2). These small discrepancies in bond lengths and angles leads to degraded performance [16, 15]. To mitigate this, we propose MOF matching, a preprocessing pro-

---

**Algorithm 1** Canonicalization of rotation targets

**Input:** Clean building block $\mathcal{C}_1 = (A, X_1)$, noisy building block $\mathcal{C}_0 = (A, X_0)$ where $X_0 = \Delta q_{1\rightarrow 0} \cdot X_1$

**Output:** Canonical rotation target $\Delta q^*_{1\rightarrow 0}$.

  1: Identify $G(\mathcal{C}_0) := \{\mathfrak{g} \in O(3) | \mathfrak{g} \cdot \mathcal{C}_0 = \mathcal{C}_0\}$
  2: Identify $\mathcal{R} := G(\mathcal{C}_0) \cap SO(3)$
  3: Solve $\mathfrak{g}^* \leftarrow \arg\min_{\mathfrak{g} \in \mathcal{R}} \mathrm{RMSD}(\mathfrak{g} \cdot X_0, X_1)$
  4: Compute $\Delta q^*_{1\rightarrow 0} \leftarrow \mathrm{Kabsch}(X_1, \mathfrak{g}^* \cdot X_0)$

---

cedure adapted from conformer matching from Jing et al. [16]. Specifically, MOF matching replaces each metal building block with its template structure from the library and each organic linker with an RDKit-generated structure whose torsion angles are optimized to closely match the original conformation. The resulting matched coordinates are used for training to ensure consistency with inference-time input. See Appendix E for details.

## 6 Experiments

We evaluate MOFFLOW-2 on two key tasks, MOF structure prediction and MOF generation, to demonstrate its ability to predict accurate 3D structures and design novel MOFs. We first describe the shared data preprocessing pipeline, then present the structure prediction task in Section 6.1 and the MOF generation task in Section 6.2. Additional experimental details are provided in Appendix F.

**Data preprocessing.** We generally follow the preprocessing pipeline from prior work [11, 10]. Starting with the dataset from Boyd et al. [36], we apply `metal-oxo` decomposition algorithm from `MOFid` [37] and discard any structures containing more than 20 building blocks [11]. The resulting dataset is split into an 8:1:1 ratio for train/valid/test sets in the structure prediction task, and into a 9.5:0.5 train/valid split for MOF generation [11, 10]. Since the dataset [36] consists of hypothetical structures, we further filter out invalid MOFs using `MOFChecker` [38], and apply the MOF matching

Table 2: **Structure prediction accuracy.** We compare the structure prediction performance of random search (RS), evolutionary algorithm (EA), DiffCSP, MOFFLOW, and MOFFLOW-2. MR is the match rate, RMSE is the root mean squared error, and $-$ indicates no match. stol is the site tolerance for matching criteria.

|  |  | RS | EA | DiffCSP [8] | | MOFFLOW [10] | | **MOFFLOW-2** | |
| --- | --- | --- | --- | --- | --- | --- | --- | --- | --- |
|  | # of samples | 20 | 20 | 1 | 5 | 1 | 5 | 1 | 5 |
| stol = 0.3 | MR (%) ↑ | 0.00 | 0.00 | 0.01 | 0.08 | 5.28 | 8.68 | 8.20 | **15.98** |
|  | RMSE (Å) ↓ | – | – | 0.1554 | **0.1299** | 0.2036 | 0.2039 | 0.1894 | 0.1842 |
| stol = 0.5 | MR (%) ↑ | 0.00 | 0.00 | 0.23 | 0.87 | 21.93 | 32.71 | 28.71 | **43.95** |
|  | RMSE (Å) ↓ | – | – | 0.3896 | 0.3982 | 0.3329 | 0.3290 | 0.3094 | **0.2925** |

procedure to address distributional shift at test time, as discussed in Section 5.3. Preprocessing details and dataset statistics are available in Appendix F.

## 6.1 MOF structure prediction

**Baselines.** We compare our model against both classic optimization- and learning-based approaches. For the classic baselines, we consider random search (RS) and the evolutionary algorithm (EA) implemented in CrySPY [39]. Among the learning-based baselines, we include DiffCSP [8], a general crystal structure prediction model, and MOFFLOW [10], a MOF-specific structure prediction model. To ensure a fair comparison, we assume that no models have access to ground-truth local coordinates. Accordingly, we retrain MOFFLOW with matched coordinates (Section 5.3) and evaluate it using the initialized structures described in Section 4.2.

**Metrics.** We use match rate (MR) and root mean square error (RMSE) computed with the StructureMatcher class from pymatgen [40]. StructureMatcher determines whether a predicted structure matches the reference based on specified tolerances. The RMSE is calculated only over the matched structures. We report results under two tolerance settings, $(0.5, 0.3, 10.0)$ and $(0.3, 0.3, 1.0)$, corresponding to site positions, fractional lengths, and lattice angle tolerances, respectively. The former is a standard threshold in CSP tasks [8, 9], while the latter is stricter for fine-grained evaluation. Note that while MOFFLOW and MOFFLOW-2 are trained on matched coordinates, we evaluate their predicted structures against ground-truth coordinates for fair comparison.

**Results.** Table 2 shows that MOFFLOW-2 outperforms all baselines across both tolerance settings. Consistent with prior work [10], we observe that the optimization-based methods (RS and EA) and the general CSP baseline (DiffCSP) achieve near-zero match rates, indicating their limited effectiveness on MOF structures. Compared to MOFFLOW [10], our method achieves higher match rates and lower RMSE, underscoring the importance of explicitly modeling torsion angles for accurate structure prediction. We also provide visualizations of property distributions in Appendix G.1.

## 6.2 MOF generation

We evaluate the generative performance of MOFFLOW-2 by (1) measuring validity, novelty, and uniqueness (VNU), and (2) comparing the distribution of MOF properties against that of the training set. We benchmark against MOFDiff [11], a coarse-grained diffusion model for MOF generation. Both models generate 10,000 samples using their respective pipelines. To isolate the effect of the generative model, no force-field relaxation is applied to the generated structures. Additional results on conditional generation and energy-level evaluation are provided in Appendix G.2 and Appendix G.3.

**Metrics.** A generated MOF is considered valid if it passes the MOFChecker test [38]. It is considered novel if its MOFid [37] does not appear in the training set. Uniqueness is computed as the proportion of distinct structures after removing duplicates from the generated set. We also report the percentage of novel building blocks (NBB), defined as the proportion of samples that are both valid and contain at least one building block not present in the training set. Additionally, we use Zeo++ [41] to evaluate MOF-specific properties, such as surface area and density.

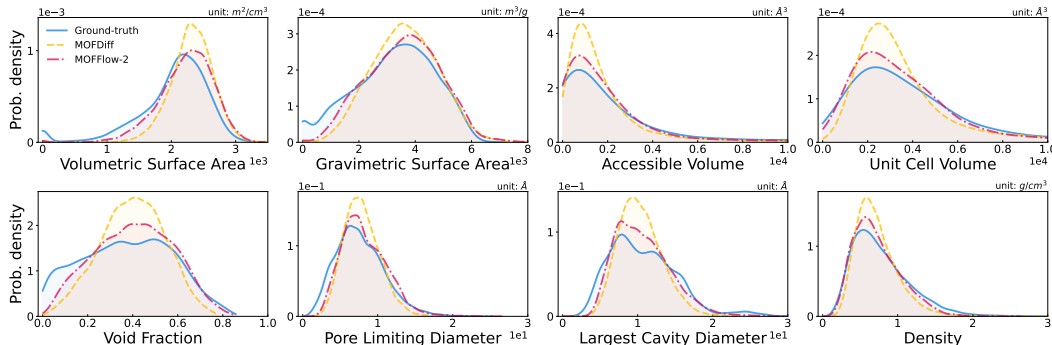

Figure 4: **Property distributions.** We compare MOF property distributions of the ground-truth, MOFDiff, and MOFFLOW-2. The distribution has been smoothed with kernel density estimation. Compared to MOFDiff, MOFFLOW-2 closely aligns with the ground-truth distribution and covers a broader range of values, demonstrating that MOFFLOW-2 can generate MOFs with diverse properties.

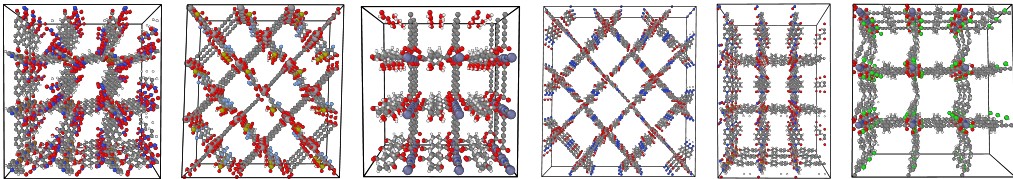

Figure 5: **Samples from MOFFLOW-2.** Visualizations of samples generated by MOFFLOW-2 that are valid, novel, and unique.

**Results.** Table 3 summarizes the generation performance of MOFFLOW-2 compared to MOFDiff. MOFFLOW-2 achieves higher scores across all metrics. Notably, MOFFLOW-2 not only generates MOFs with novel combinations of known building blocks but also produces entirely new building blocks, demonstrating its potential for discovering previously unseen MOFs. In addition, MOFFLOW-2 is also faster than MOFDiff, which depends on an optimization-based self-assembly procedure for final structure construction. As shown in Figure 4, the property distribution of MOFFLOW-2 aligns more closely with the ground-truth. Importantly, it spans a broader range of values than MOFD-iff, indicating that MOFFLOW-2 generates MOFs with more diverse physical properties. We provide visualizations of representative VNU samples generated by MOFFLOW-2 in Figure 5.

Table 3: **MOF generation results.** MOFFLOW-2 outperforms MOFDiff in validity, novelty, uniqueness (VNU), and average sampling time. It can also generate MOFs with novel building blocks (NBB).

|       |       | MOFDiff | MOFFLOW-2 |
|-------|-------|---------|-----------|
| Valid | (%) ↑ | 10.13   | **38.84** |
| VNU   | (%) ↑ | 7.95    | **31.35** |
| NBB   | (%) ↑ | 0.00    | **10.10** |
| Time  | (s) ↓ | 3.19    | **1.82**  |

## 7 Conclusion

We introduced MOFFLOW-2, a two-stage generative model for MOF design and structure prediction. In the first stage, the building block generator designs MOF sequences in SMILES, which are initialized into 3D structures using a predefined metal library and RDKit [14]. In the second stage, the structure prediction module assembles the complete MOF by jointly predicting rotations, translations, torsions, and lattice parameters. Experimental results show that MOFFLOW-2 outperforms existing models in both generative design and structure prediction.

Despite its strong performance, MOFFLOW-2 still has several limitations, including dependence on RDKit for conformer initialization, partial conditioning in property-guided generation (Appendix G.2), and approximate energy evaluation with machine learning interatomic potentials (Appendix G.3). We provide a detailed discussion of these limitations in Appendix H.

## Acknowledgements

This work was supported by the NVIDIA Academic Grant Program. We thank NVIDIA for providing the GPU resources used in this work. This work was also supported by Basic Science Research Program through the National Research Foundation of Korea(NRF) funded by the Ministry of Education (RS-2025-25435147); Institute for Information & communications Technology Planning & Evaluation(IITP) grant funded by the Korea government(MSIT) (RS-2019-II190075, Artificial Intelligence Graduate School Support Program(KAIST)); National Research Foundation of Korea(NRF) grant funded by the Ministry of Science and ICT(MSIT) (No. RS-2022-NR072184), GRDC(Global Research Development Center) Cooperative Hub Program through the National Research Foundation of Korea(NRF) grant funded by the Ministry of Science and ICT(MSIT) (No. RS-2024-00436165); and Institute of Information & Communications Technology Planning & Evaluation(IITP) grant funded by the Korea government(MSIT) (RS-2025-02304967, AI Star Fellowship(KAIST)).

We thank Vignesh Ram Somnath for his help with model scaling. We also thank Seonghyun Park, Hyomin Kim, and Junwoo Yoon for their help improving the manuscript.

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

# A Implementation details for building block generator

We provide additional details for implementing the building block generator (Section 4.1), including the tokenization process, batching strategy, and hyperparameter settings.

## A.1 Example of MOF sequence tokenization.

We illustrate the tokenization process for a canonical MOF sequence defined in Section 4.1, which transforms the string into a sequence of discrete symbols $[b_1, \ldots, b_S]$.

Consider the following canonicalized 2D MOF sequence $\mathcal{B}_{2D}$.

```
<BOS> [Cu+][Cu+] <SEP> c1cnccn1.O=C([O-])c1ccc(C(=O)[O-])cc1 <EOS>
```

We apply the SMILES tokenization regex from Schwaller et al. [28] to split the string into the following tokens.

```
<BOS>, [Cu+], [Cu+], <SEP>, c, 1, c, n, c, c, n, 1, ., O, =, C,
(, [O-], ), c, 1, c, c, c, (, C, (, =, O, ), [O-], ), c, c, 1, <EOS>
```

We then map each token to a unique vocabulary index to get $[b_1, \ldots, b_S]$.

```
2, 7, 7, 4, 8, 9, 8, 10, 8, 8, 10, 9, 11, 12, 13, 14, 15, 16,
17, 8, 9, 8, 8, 8, 15, 14, 15, 13, 12, 17, 16, 17, 8, 8, 9, 3
```

## A.2 Training details.

**Batching.** During training, we use dynamic batching [42] to prevent out-of-memory errors and reduce padding inefficiencies caused by highly variable sequence lengths. The key idea is to (1) limit each batch to a fixed maximum number of tokens and (2) group sequences of similar lengths. Specifically, we add the samples to a heap-based buffer sorted by sequence length. Once the buffer exceeds a predefined capacity, we add the longest sequences to the current batch until reaching the token limit, then start a new batch.

**Hyperparameters.** In Table 4, we present the hyperparameters for training the building block generator, including the batch configuration, model architecture, and optimizer settings.

**Codebase.** We implemented our sequence model with `x-transformers` [43]. We appreciate the author for the open-source implementation.

| Hyperparameter | Value |
|---|---|
| Max tokens | 8000 |
| Number of layers | 6 |
| Hidden dimension | 1024 |
| Number of heads | 8 |
| Rotary positional embedding | True |
| Flash attention | True |
| Scale normalization | True |
| Optimizer | AdamW |
| Learning rate | 3e-4 |
| Betas | (0.9, 0.999) |
| Weight decay | 0.0 |
| Epochs | 20 |

Table 4: Hyperparameters for training the building block generator.

# B  Analysis and visualization of metal building blocks

Figure 6a and Figure 6b show the RMSD histograms of the extracted metal building blocks for the generation and structure prediction tasks, respectively. There are only 7 and 8 distinct metal building block types, despite the large sizes of the training datasets (>150k). The RMSD histograms further indicate that all metal types exhibit low structural variability. These observations justify our template-based strategy for initializing the metal structures, described in Section 4.2. We also provide visualizations of the metal building blocks in Figure 7.

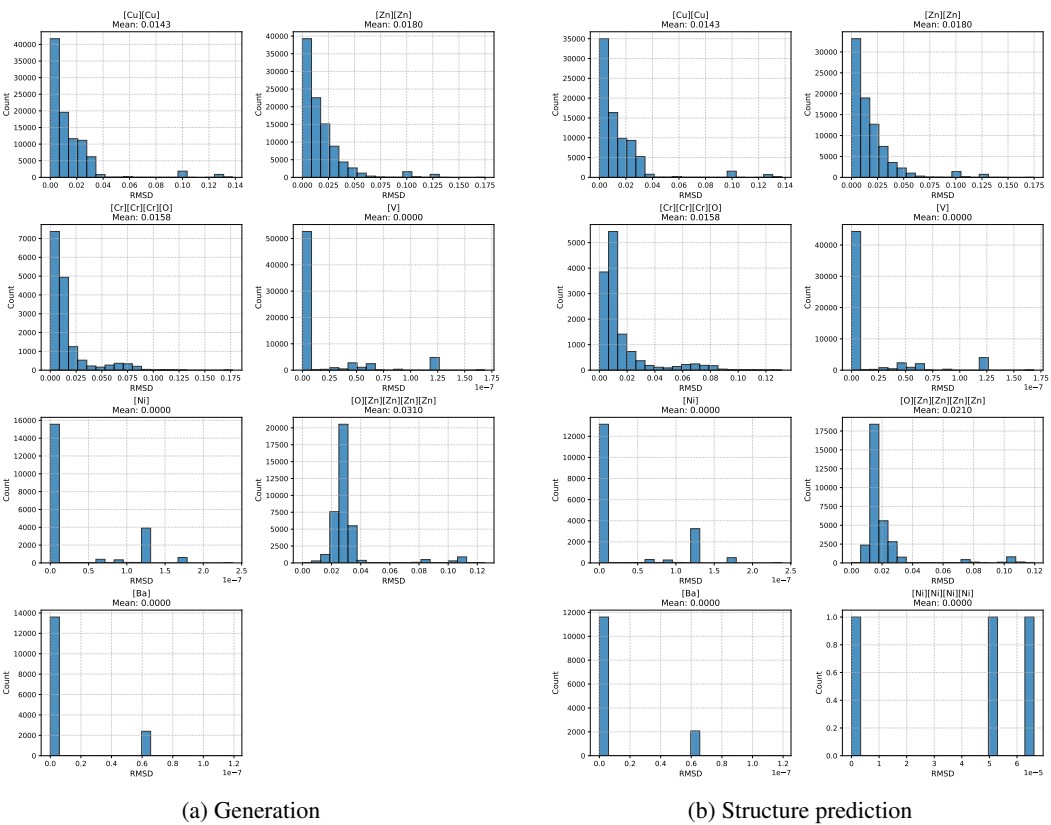

(a) Generation                                    (b) Structure prediction

Figure 6: **Metal RMSD histograms.** RMSD histograms of the metal building blocks for (a) generation and (b) structure prediction task, respectively. The low diversity and structural variability support our template-based approach for metal structure initialization.

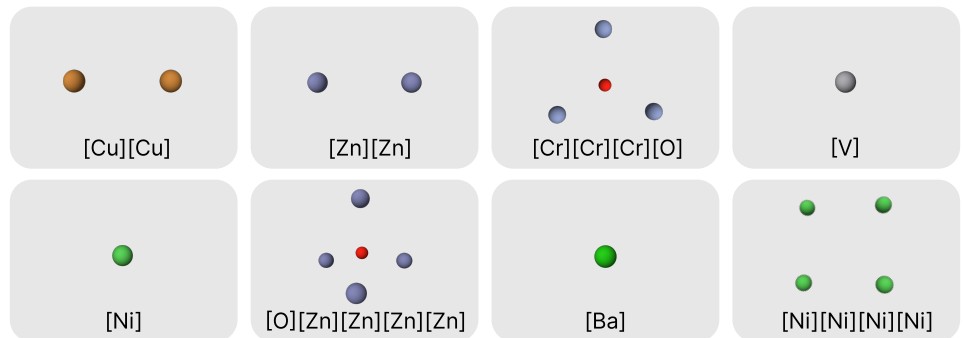

Figure 7: Visualizations of metal building blocks extracted from the training dataset.

# C Algorithmic details for structure prediction model

## C.1 Conditional flows and conditional vector fields

Here, we derive the conditional flow $z_t = \exp_{z_0}(t \log_t(z_t))$ and conditional vector field $u_t(z_t|z_1) = \log_{z_t}(z_1)/(1-t)$ for each structural component: rotations, translations, torsions, and lattice parameters.

**Translation and lattice parameters.** The translations $\tau \in \mathbb{R}^{M \times 3}$ and lattice parameters $\ell = (a, b, c, \alpha, \beta, \gamma) \in \mathbb{R}_+^3 \times [0°, 180°]^3$ lie on the Euclidean space, where the exponential and logarithmic maps are $\exp_a b = a + b$ and $\log_a b = b - a$. Therefore, the conditional flow can be written as:

$$\tau_t = (1-t)\tau_0 + t\tau_1, \quad \ell_t = (1-t)\ell_0 + t\ell_1, \tag{8}$$

and the conditional vector fields as:

$$u_t(\tau_t|\tau_1) = \frac{\tau_1 - \tau_t}{1-t}, \quad u_t(\ell_t|\ell_1) = \frac{\ell_1 - \ell_t}{1-t}. \tag{9}$$

**Torsions.** Each torsion angle $\phi \in [-\pi, \pi)$ lies on a torus $\mathbb{T}$, where the exponential and logarithmic maps are defined as $\exp_a b = \text{wrap}(a + b)$, $\log_a b = \text{wrap}(b - a)$ with $\text{wrap}(x) = (x + \pi) \bmod (2\pi) - \pi$ [44]. The corresponding conditional flow and conditional vector field are:

$$\phi_t = \text{wrap}(t \cdot \text{wrap}(\phi_1 - \phi_0) + \phi_0), \quad u_t(\phi_t|\phi_1) = \frac{\text{wrap}(\phi_1 - \phi_t)}{1-t}. \tag{10}$$

**Rotations.** Each rotation $q$ lie on $SO(3)$, where we define $\log_a b = \log(ba^{-1})$ and $\exp_a b = \exp(b)a$. Then, the conditional flow is given by

$$q_t = \exp_{q_1}((1-t)\log_{q_1}(q_1)) = \exp((1-t)\log(q_0 q_1^{-1}))q_1, \tag{11}$$

where $\exp : \mathfrak{so}(3) \to SO(3)$ and $\log : SO(3) \to \mathfrak{so}(3)$ are the exponential and logarithmic maps for $SO(3)$, respectively [45]. Since we target $\Delta q_{t \to 1} := q_t q_1^\top$ (Section 5.1),

$$\Delta q_{t \to 1} = \exp((1-t)\log(\Delta q_{1 \to 0})), \quad \Delta q_{1 \to 0} \sim \mathcal{U}(SO(3)). \tag{12}$$

Accordingly, the conditional vector field is given by

$$u_t(q_t|q_1) = \frac{\log q_1 q_t^{-1}}{1-t} = \frac{\log \Delta q_{t \to 1}}{1-t}, \tag{13}$$

where $\Delta q_{t \to 1} = \Delta q_{1 \to t}^\top = q_1 q_t^\top$.

## C.2 Applying rotation, translation, and torsion

**Rotations and translations.** Given a MOF structure $\mathcal{S} = (A, X, \ell)$ with coordinates decomposed as $X = [X^{(1)}, \dots, X^{(M)}]$, we apply rotation $\Delta q = (\Delta q^{(1)}, \dots, \Delta q^{(M)}) \in SO(3)^M$ and translation $\tau = (\tau^{(1)}, \dots, \tau^{(M)}) \in \mathbb{R}^{M \times 3}$ independently to each building block:

$$(\Delta q, \tau) \cdot X = [X^{(m)} \Delta q^{(m)\top} + 1_{N_m} \tau^{(m)}]_{m=1}^M \tag{14}$$

where $1_{N_m} = [1, \dots, 1]^\top$.

**Torsions.** We adapt the definition from Jing et al. [16] to apply torsion angles to MOF structures. Given coordinates $X = [X^{(1)}, \dots, X^{(M)}]$ and torsion angles $\phi = [\phi^{(m)} \in \mathbb{T}^{P_m}]_{m=1}^M$, where $P_m$ is the number of rotatable bonds in the $m$-th building block, we update the coordinates around each rotatable bond $(j, k)$ with current angle $\phi$ and target angle $\phi'$ as:

$$X'_{\mathcal{V}(j)} = (X_{\mathcal{V}(j)} - x_k) \exp\left((\phi' - \phi)\frac{x_j - x_k}{\|x_j - x_k\|}\right) + x_k \tag{15}$$

where $\mathcal{V}(j)$ denotes atoms on the side of atom $j$ and $\exp : \mathfrak{so}(3) \to SO(3)$. The other side remains unchanged as $X'_{\mathcal{V}(k)} = X_{\mathcal{V}(k)}$.

## C.3 Training and inference algorithm

Algorithm 2 and Algorithm 3 present the training and inference algorithms for the structure prediction model, respectively. During training, the model outputs rotations $\Delta\hat{Q}_{t\to1} \in \mathbb{R}^{M\times3\times3}$ and torsions $\hat{\Phi}_1 \in \mathbb{R}^{P\times2}$ in the Euclidean space, which are directly supervised with the targets $\Delta q_{t\to1}$ and $\phi_1$. At inference, these outputs are projected onto their respective manifolds as $\Delta\hat{q}_{t\to1} = \text{Procrustes}(\Delta\hat{Q}_{t\to1}) \in SO(3)^M$ and $\hat{\phi}_1 = \hat{\Phi}_1/\|\hat{\Phi}_1\|_2 \in \mathbb{T}^P$.

---

**Algorithm 2** Training algorithm

---

**Input:** Training dataset $\{(\mathcal{B}_{2D}, \mathcal{S}_1)\}$ where $\mathcal{B}_{2D}$ is 2D building block representation and $\mathcal{S}_1 = (A, X_1, \ell_1)$ is corresponding 3D MOF structure.

1: **for** $(\mathcal{B}_{2D}, \mathcal{S}_1)$ **do**
2:     Sample time $t \sim \mathcal{U}(0, 1)$.
3:     Sample $\Delta q_{1\to0}, \tau_0, \phi_0, \ell_0$ from prior distributions defined in Section 5.1.
4:     Interpolate $\Delta q_{1\to t}, \tau_t, \phi_t, \ell_t$ according to Appendix C.1
5:     Apply $X_t \leftarrow (\Delta q_{1\to t}, \tau_t, \phi_t) \cdot X_1$ according to Appendix C.2.
6:     Compute outputs $(\Delta\hat{Q}_{t\to1}, \hat{\tau}_1, \hat{\Phi}_1, \hat{\ell}_1) = \mathcal{F}_\theta(A, X_t, \ell_t)$.
7:     Optimize loss $\mathcal{L}(\theta)$ in Equation (5).
8: **end for**

---

**Algorithm 3** Inference algorithm

---

**Input:** 2D building block representation $\mathcal{B}_{2D}$, number of integration steps $T$
**Output:** Predicted MOF structure $(A, X_1, \ell_1)$

1: Initialize structure $\tilde{X}_0 \leftarrow \text{Initialize}(\mathcal{B}_{2D})$ according to Section 4.2.
2: Sample $\Delta q_0, \tau_0, \phi_0, \ell_0$ from prior distributions defined in Section 5.1.
3: Apply transformation $X_0 \leftarrow (\Delta q_0, \tau_0, \phi_0) \cdot \tilde{X}_0$.
4: Set $\Delta t \leftarrow 1/T$.
5: **for** $i = 0, \ldots, T-1$ **do**
6:     Set $t \leftarrow i/T$.
7:     Predict $(\Delta Q_{t\to1}, \tau_1, \Phi_1, \ell_1) = \mathcal{F}_\theta(A, X_t, \ell_t)$.
8:     Project to manifold $\Delta q_{t\to1} \leftarrow \text{Procrustes}(\Delta Q_{t\to1})$, $\phi_1 \leftarrow \Phi_1/\|\Phi_1\|_2$.
9:     Take a step $\Delta q_{t\to t+\Delta t} \leftarrow \text{EulerStep}(\Delta q_{t\to1}; t, \Delta t)$.
10:     Take a step $y_{t+\Delta t} \leftarrow \text{EulerStep}(y_t, y_1, \Delta t)$ for $y \in \{\tau, \phi, \ell\}$.
11:     Apply transformation $X_{t+\Delta t} \leftarrow (\Delta q_{t\to t+\Delta t}, \tau_{t+\Delta t}, \phi_{t+\Delta t}) \cdot X_t$.
12: **end for**

---

# D  Model architecture for structure prediction

## D.1  Initialization module

**Building block index.** We define the building block index $k$ as introduced in Section 5.2 to resolve ambiguity between building blocks of the same type – i.e., those sharing the same SMILES representation. To assign consistent indices, we apply the following lexicographic ordering rules:

1. Metal building blocks are always indexed before organic building blocks.
2. Within each group (metal or organic), blocks are ordered by molecular weight.
3. Ties between building blocks of the same type are resolved by sorting their centroid coordinates $(x, y, z)$ in ascending order of $x$, then $y$, then $z$ [46].

## D.2  Interaction module

The interaction module follows a Transformer encoder architecture [47] with root mean square layer normalization [RMSNorm; 48] (Figure 2). Here, we provide further implementation details, including the model architecture (Algorithm 4) and associated hyperparameters (Table 5).

---

**Algorithm 4** Interaction module (Transformer encoder)

---

**Input:** Initialized atom embeddings $H = [h_i \in \mathbb{R}^D]_{i=1}^N$, bond edges $\mathcal{E}_{\text{bond}}$, cutoff radius $c \in \mathbb{R}_+$, maximum number of neighbors $N_{\text{max}}$.
**Output:** Updated atom embeddings $H' = [h_i' \in \mathbb{R}^D]_{i=1}^N$.

1: # *Construct edges and edge features [15]*
2: Construct radius edges $\mathcal{E}_{\text{radius}} \leftarrow \{(i, j) \mid \|x_i - x_j\|_2 < c\}$ with $|\mathcal{N}(i)| \leq N_{\text{max}}, \forall i$.
3: Combine edges $\mathcal{E} \leftarrow \mathcal{E}_{\text{radius}} \cup \mathcal{E}_{\text{bond}}$.
4: Construct edge features $E = [e_{ij}]_{(i,j) \in \mathcal{E}}$ with $e_{ij} = [b(i, j), \text{RBF}(\|x_i - x_j\|_2)]$.
5: # *Update node features*
6: $H \leftarrow H + \text{MHA}(\text{RMSNorm}(H), E, \mathcal{E})$       ▷ Refer to Section 5.2 for details on MHA.
7: $H' \leftarrow H + \text{FFN}(\text{RMSNorm}(H))$

---

| Hyperparameter | Value |
|---|---|
| Number of layers | 10 |
| Maximum radius $c$ | 50 |
| Maximum neighbors $N_{\text{max}}$ | 130 |
| Node embedding dimension | 1024 |
| RBF embedding dimension | 128 |
| Number of heads | 16 |
| FFN embedding dimension | 4096 |

Table 5: Hyperparameters for the interaction module.

### D.3 Output module

We describe the architectures of the four prediction heads in the output module (Section 5.2). We begin with the block attention pooling module, which aggregates atom-level node embeddings from the interaction module into block-level embeddings (Algorithm 5). We then detail the architectures of the rotation (Algorithm 6), translation (Algorithm 7), lattice (Algorithm 8), and torsion heads (Algorithm 9). The corresponding hyperparameters are summarized in Table 6.

---

**Algorithm 5** Block attention pooling module (BlockAttentionPool)

---

**Input:** Atom embeddings from interaction module $H = [h_i \in \mathbb{R}^D]_{i=1}^N$.

**Output:** Block-wise embeddings $H_{bb} = [h_{bb}^{(m)} \in \mathbb{R}^D]_{m=1}^M$ for each building block.

1: # *Construct building block coordinates* $[x_{bb}^{(m)}]_{m=1}^M \in \mathbb{R}^{M \times 3}$ *and features* $[h_{bb}^{(m)}]_{m=1}^M \in \mathbb{R}^{M \times D}$.
2: **for** $\forall m = 1, \ldots, M$ **do**
3:      Compute building block centroid $x_{bb}^{(m)} \leftarrow \frac{1}{N_m} \sum_{i \in \mathcal{V}^{(m)}} x_i$.
4:      Compute averaged feature $h_{bb}^{(m)} \leftarrow \frac{1}{N_m} \sum_{i \in \mathcal{V}^{(m)}} h_i$.
5: **end for**
6: # *Construct edges and edge features*
7: Construct edges within the building blocks $\mathcal{E}_{bb} \leftarrow \{(m, j) \in [M] \times \mathcal{V}^{(m)}\}$.
8: Construct edge distance features $[e_{mj}]_{(m,j) \in \mathcal{E}_{bb}}$ where $e_{mj} = \mathrm{RBF}(\|x_{bb}^{(m)} - x_j\|_2)$.
9: # *Attention*
10: **for** $\forall m = 1, \ldots, M$ **do**
11:      Compute query $q_m \leftarrow \mathrm{Linear}_Q(h_{bb}^{(m)})$.
12:      Compute key $k_{mj} \leftarrow \mathrm{Linear}_K([h_{bb}^{(m)}, e_{mj}]), \forall j \in \mathcal{V}^{(m)}$.
13:      Compute value $v_{mj} \leftarrow \mathrm{Linear}_V([h_{bb}^{(m)}, e_{mj}]), \forall j \in \mathcal{V}^{(m)}$.
14:      Compute attention score $a_{mj} \leftarrow \mathrm{Softmax}_{j \in \mathcal{V}^{(m)}}(q_m^\top k_{mj} / \sqrt{D})$.
15:      Aggregate and update $h_{bb}^{(m)} \leftarrow \mathrm{Linear}(\sum_{j \in \mathcal{V}^{(m)}} a_{mj} v_{mj})$.
16: **end for**

---

**Algorithm 6** Rotation head

---

**Input:** Atom embeddings from interaction module $H = [h_i \in \mathbb{R}^D]_{i=1}^N$.

**Output:** Rotation predictions for each block $\Delta \boldsymbol{q}_{t \to 1} \in SO(3)^M$

1: Block-wise attention pooling $H_{bb} \leftarrow \mathrm{BlockAttentionPool}(H)$.
2: Raw rotation output $\Delta \boldsymbol{Q}_{t \to 1} \leftarrow \mathrm{Linear}(\mathrm{GELU}(\mathrm{Linear}(H_M)))$.
3: **if** is inference **then**
4:      Project $\Delta \boldsymbol{q}_{t \to 1} \leftarrow \mathrm{Procrustes}(\Delta \boldsymbol{Q}_{t \to 1})$.
5: **end if**

---

**Algorithm 7** Translation head

---

**Input:** Atom embeddings from interaction module $H = [h_i \in \mathbb{R}^D]_{i=1}^N$.

**Output:** Translation predictions for each block $\boldsymbol{\tau}_1 \in \mathbb{R}^{M \times 3}$

1: Block-wise attention pooling $H_{bb} \leftarrow \mathrm{BlockAttentionPool}(H)$.
2: Update with MLP $\tilde{\boldsymbol{\tau}}_1 \leftarrow \mathrm{Linear}(\mathrm{GELU}(\mathrm{Linear}(H_M)))$.
3: Remove mean $\boldsymbol{\tau}_1 \leftarrow \tilde{\boldsymbol{\tau}}_1 - \sum_{m=1}^M \tilde{\tau}_1^{(m)}$.

---

**Algorithm 8** Lattice head

---

**Input:** Atom embeddings from interaction module $H = [h_i \in \mathbb{R}^D]_{i=1}^N$.

**Output:** Lattice prediction $\boldsymbol{\ell}_1 \in \mathbb{R}_+^3 \times [0°, 180°]$.

1: Block-wise attention pooling $H_{bb} \leftarrow \mathrm{BlockAttentionPool}(H)$.
2: Average and MLP $\tilde{\boldsymbol{\ell}}_1 \leftarrow \mathrm{MLP}(\sum_{m=1}^M h_{bb}^{(m)})$.      ▷ Two-layer MLP with GELU activation.
3: Apply $\boldsymbol{\ell}_1 \leftarrow \mathrm{Softplus}(\tilde{\boldsymbol{\ell}}_1)$.

---

---

**Algorithm 9** Torsion head

---

**Input:** Atom embeddings from interaction module $H = [h_i \in \mathbb{R}^D]_{i=1}^N$, rotatable bond index $\{(i_p, j_p, k_p, l_p)\}_{p=1}^P$, cutoff radius $c \in \mathbb{R}_+$, maximum number of neighbors $N_{\max}$.
**Output:** Torsion predictions $\phi_1 \in \mathbb{T}^P$.

1: # *Construct rotatable bond coordinates* $[x_{rot}^{(p)}]_{p=1}^P \in \mathbb{R}^{P \times 3}$ *and features* $[h_{rot}^{(p)}]_{p=1}^P \in \mathbb{R}^{P \times D}$.
2: **for** $\forall p = 1, \ldots, P$ (i.e., each rotatable bond) **do**
3:      # *For clarity let* $(i, j, k, l) \leftarrow (i_p, j_p, k_p, l_p)$.
4:      Compute rotatable bond centroid $x_{\text{rot}}^{(p)} \leftarrow (x_j + x_k)/2$.
5:      Compute rotatable bond feature $h_{\text{rot}}^{(p)} \leftarrow \text{Linear}([h_i, h_j, h_k, h_l]) + \text{Linear}([h_l, h_k, h_j, h_i])$.
6: **end for**
7: # *Construct edges and edge features*
8: Construct edges $\mathcal{E}_{\text{rot}} \leftarrow \{(p, j) \in [P] \times [N] | \|x_{\text{rot}}^{(p)} - x_j\|_2 < c\}$ with $|\mathcal{N}(p)| \leq N_{\max}, \forall p$.
9: Construct edge distance features $[e_{pj}]_{(p,j) \in \mathcal{E}_{\text{rot}}}$ where $e_{pj} = \text{RBF}(\|x_{\text{rot}}^{(p)} - x_j\|_2)$.
10: # *Attention*
11: **for** $\forall p = 1, \ldots, P$ **do**
12:      Compute query $q_p \leftarrow \text{Linear}_Q(h_{\text{rot}}^{(p)})$.
13:      Compute key $k_{pj} \leftarrow \text{Linear}_K([h_{\text{rot}}^{(p)}, e_{pj}]), \forall (p, j) \in \mathcal{E}_{\text{rot}}$.
14:      Compute value $v_{pj} \leftarrow \text{Linear}_V([h_{\text{rot}}^{(p)}, e_{pj}]), \forall (p, j) \in \mathcal{E}_{\text{rot}}$.
15:      Compute attention score $a_{pj} \leftarrow \text{Softmax}_{j \in \mathcal{N}(p)}(q_p^\top k_{pj}/\sqrt{D})$.
16:      Aggregate and update $h_{\text{rot}}^{(p)} \leftarrow \text{Linear}(\sum_{j \in \mathcal{N}(p)} a_{pj} v_{pj})$.
17: **end for**
18: Apply $\mathbf{\Phi}_1 \leftarrow \text{MLP}(H_{\text{rot}})$ where $H_{\text{rot}} = [h_{\text{rot}}^{(p)}]_{p=1}^P$.      ▷ Two-layer MLP with GELU activation.
19: **if** is inference **then**
20:      Project $\phi_1 \leftarrow \mathbf{\Phi}_1/\|\mathbf{\Phi}_1\|_2$.
21: **end if**

| Hyperparameter | Value |
|---|---|
| Node embedding dimension | 1024 |
| RBF embedding dimension | 128 |
| Number of heads for BlockAttentionPool | 16 |
| Maximum radius ($c$) for TorsionHead | 5 |
| Maximum neighbors ($N_{\max}$) for TorsionHead | 24 |

Table 6: Hyperparameters for the output module.

# E  MOF matching

We describe the MOF matching procedure, a preprocessing step used to reduce distributional shift between training and inference, as outlined in Section 5.3. Given an MOF structure and its building blocks, we first determine whether each block contains metal elements to classify it as metal or organic. For metal blocks, we retrieve the corresponding template from the metal library and align it to the ground-truth structure. For organic blocks, we initialize the structure using RDKit, optimize torsion angles via differential evolution to minimize RMSD with the ground-truth [16], and then align the result. After processing all blocks, we compute the RMSD between the reconstructed and ground-truth structures. This procedure is repeated three times, and structures with final RMSD below 0.5Å are retained. The full algorithm for a single matching iteration is shown in Algorithm 10.

---

**Algorithm 10** An iteration of MOF matching

---

**Input:**  DFT-relaxed MOF structure $\mathcal{S} = (\boldsymbol{A}, \boldsymbol{X}, \boldsymbol{\ell})$, trial number $n$, base population size $p_0$, base maximum iteration $t_0$.
**Output:**  Matched MOF structure $\tilde{\mathcal{S}} = (\boldsymbol{A}, \tilde{\boldsymbol{X}}, \boldsymbol{\ell})$, $\mathrm{RMSD}(\mathcal{S}, \tilde{\mathcal{S}}) \in \mathbb{R}_+$.

1: **for** $\forall m = 1, \ldots, M$ (i.e., each building block) **do**
2:     **if** $A^{(m)}$ contains metal element **then**
3:         *# Metal building block*
4:         Retrieve the corresponding template structure $\tilde{X}^{(m)}$ from metal library (Section 4.2).
5:         Align to original coordinates $\tilde{X}^{(m)} \leftarrow \mathrm{Align}(\tilde{X}^{(m)}, X^{(m)})$.
6:     **else**
7:         *# Organic building block [16]*
8:         Initialize structure $\tilde{X}^{(m)}$ with RDKit (Section 4.2).
9:         Set parameters $p \leftarrow p_0 + 10n$, $t \leftarrow t_0 + 10n$.
10:       Optimize torsion angles $\tilde{X}^{(m)} \leftarrow \mathrm{DifferentialEvolution}(\tilde{X}^{(m)}, X^{(m)}, p, t)$.
11:       Align to original coordinates $\tilde{X}^{(m)} \leftarrow \mathrm{Align}(\tilde{X}^{(m)}, X^{(m)})$.
12:     **end if**
13: **end for**
14: Compute $\mathrm{RMSD}(\mathcal{S}, \hat{\mathcal{S}})$

---

# F   Experimental details

**Data preprocessing.** We detail the data preprocessing steps described in Section 6. First, we discard MOF structures with more than 20 building blocks since large structures that are too large may be difficult to synthesize [11]. Next, we extract key features from each structure, including Cartesian coordinates, RDKit-derived atomic features, Niggli-reduced cells, symmetrically equivalent coordinates for each building block (Section 5.3), and canonical atoms defining the torsion angles (Section 5.3). We then construct the metal library (Section 4.2) and perform MOF matching (Appendix E) on the training dataset. Finally, since the dataset is synthetic [36], we filter out invalid MOFs using `MOFChecker` [38], which ensures the presence of key elements (e.g., C, H, and a metal), no atomic overlaps or coordination issues, sufficient porosity, and the absence of highly charged fragments or isolated molecules.

**Data statistics.** We present the data statistics for structure prediction (Section 6.1) in Tables 7 to 9 and those for generation (Section 6.2) in Tables 10 and 11.

| **Property** (number of samples $= 157,474$) | **Min** | **Mean** | **Max** |
|---|---|---|---|
| number of species / atoms | 4 / 22 | 5.3 / 125.5 | 8 / 1124 |
| volume [Å$^3$] | 534.5 | 4415.5 | 104490.5 |
| density [atoms/Å$^3$] | 0.1133 | 0.7444 | 3.1651 |
| lattice $a, b, c$ [Å] | 6.86 / 8.27 / 8.56 | 12.94 / 15.61 / 19.34 | 47.00 / 47.15 / 60.81 |
| lattice $\alpha, \beta, \gamma$ [°] | 59.98 / 59.98 / 59.99 | 91.31 / 91.35 / 90.90 | 120.01 / 120.01 / 120.02 |

Table 7: Statistics of the train split for structure prediction.

| **Property** (number of samples $= 19,603$) | **Min** | **Mean** | **Max** |
|---|---|---|---|
| number of species / atoms | 4 / 22 | 5.3 / 125.9 | 8 / 1,404 |
| volume [Å$^3$] | 534.6 | 4409.4 | 105417.3 |
| density [atoms/Å$^3$] | 0.1074 | 0.7454 | 2.8989 |
| lattice $a, b, c$ [Å] | 6.86 / 8.43 / 8.57 | 12.94 / 15.64 / 19.30 | 47.24 / 47.24 / 60.62 |
| lattice $\alpha, \beta, \gamma$ [°] | 60.00 / 60.00 / 59.99 | 91.32 / 91.31 / 90.91 | 120.00 / 120.01 / 120.01 |

Table 8: Statistics of the validation split for structure prediction.

| **Property** (number of samples $= 19,792$) | **Min** | **Mean** | **Max** |
|---|---|---|---|
| number of species / atoms | 4 / 24 | 5.3 / 124.2 | 8 / 1012 |
| volume [Å$^3$] | 536.4 | 4384.2 | 123788.7 |
| density [atoms/Å$^3$] | 0.1080 | 0.7444 | 3.0982 |
| lattice $a, b, c$ [Å] | 6.86 / 8.34 / 8.57 | 12.90 / 15.56 / 19.28 | 47.14 / 47.14 / 60.95 |
| lattice $\alpha, \beta, \gamma$ [°] | 60.00 / 60.00 / 60.00 | 91.20 / 91.20 / 90.86 | 120.00 / 120.00 / 120.02 |

Table 9: Statistics of the test split for structure prediction.

| **Property** (number of samples $= 187,047$) | **Min** | **Mean** | **Max** |
|---|---|---|---|
| number of species / atoms | 4 / 22 | 5.3 / 125.4 | 8 / 1404 |
| volume [Å$^3$] | 534.5 | 4410.2 | 123788.7 |
| density [atoms/Å$^3$] | 0.1074 | 0.7445 | 3.0982 |
| lattice $a, b, c$ [Å] | 6.86 / 8.27 / 8.56 | 12.94 / 15.61 / 19.33 | 47.24 / 47.24 / 60.95 |
| lattice $\alpha, \beta, \gamma$ [°] | 59.99 / 59.98 / 59.99 | 91.29 / 91.33 / 90.88 | 120.01 / 120.01 / 120.02 |

Table 10: Statistics of the train split for generation.

| **Property** (number of samples $= 10,243$) | **Min** | **Mean** | **Max** |
|---|---|---|---|
| number of species / atoms | 4 / 26 | 5.3 / 127.6 | 8 / 980 |
| volume [Å$^3$] | 582.2 | 4481.2 | 71167.7 |
| density [atoms/Å$^3$] | 0.1074 | 0.7454 | 2.8989 |
| lattice $a, b, c$ [Å] | 6.87 / 8.47 / 8.58 | 12.91 / 15.69 / 19.42 | 38.87 / 39.36 / 60.21 |
| lattice $\alpha, \beta, \gamma$ [°] | 60.00 / 60.00 / 60.00 | 91.26 / 91.20 / 90.80 | 120.00 / 119.99 / 120.00 |

Table 11: Statistics of the validation split for generation.

### F.1 Training details

**Baselines.** We follow Kim et al. [10] to RS and EA, using `CrySPY`[49] with CHGNet[50] for energy-based optimization. RS employs symmetry-based structure generation. For EA, we start with 5 random structures, select 4 parents via tournament selection, and generate offspring using 10 crossovers, 4 permutations, 2 strains, and 2 elites, iterating up to 20 generations.

For DiffCSP, we use a radius cutoff of 5Å and a batch size of 64. The model was trained for 500 epochs on an 80GB NVIDIA A100 GPU, taking approximately 5 days. Inference is performed with 1000 steps. All other settings follow the defaults in Jiao et al. [8]. FlowMM [9] is excluded from the baselines due to its high memory demands; based on our estimates, training it for 500 epochs would take over 30 days on an 80GB A100 GPU, which exceeds our resource constraints.

**MOFFLOW-2.** We train our model on 8 80GB A100 GPUs for 200 epochs (about 4 days). To avoid out-of-memory issues caused by the large variation in MOF sizes, we use a dynamic batching strategy that limits each batch to a maximum of 1500 atoms. Specifically, the samples are first added to a buffer and, once the buffer exceeds a predefined capacity, they are added to the batch in a first-in-first-out manner until the maximum limit is reached. We use AdamW optimizer [51] with a learning rate of 1e-5, betas (0.9, 0.98), and no weight decay. Inference is performed in 50 steps.

## G Additional experiments

### G.1 Property evaluation for structure prediction

Figure 8 presents the property distributions for the structure prediction task (Section 6.1). All baselines closely align with the ground-truth, indicating that key MOF properties are well preserved.

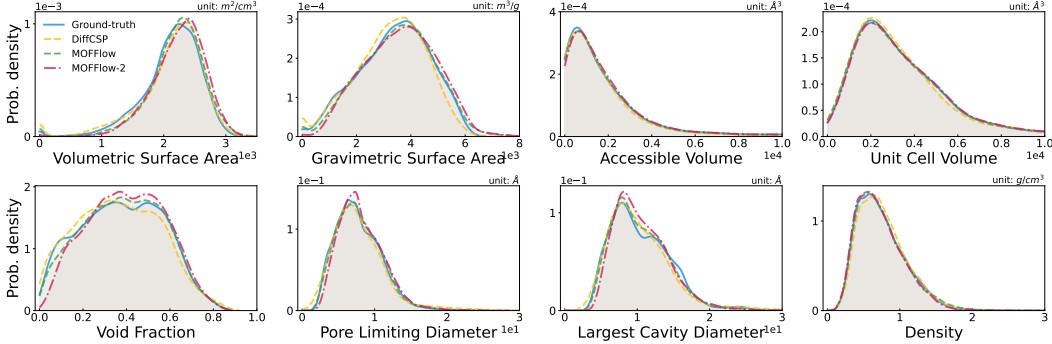

Figure 8: Key property distributions of MOFs generated by DiffCSP, MOFFLOW, and MOFFLOW-2. Distributions are smoothed using kernel density estimation. All baseline methods closely match the reference data distribution.

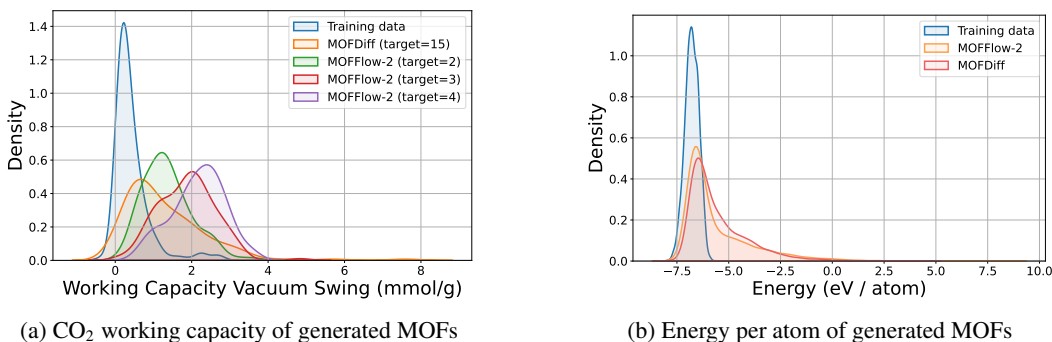

(a) $CO_2$ working capacity of generated MOFs

(b) Energy per atom of generated MOFs

Figure 9: (a) $CO_2$ working capacity distribution of MOFs generated by MOFFLOW-2 and MOFDiff, evaluated with GCMC simulations. (b) Energy per atom distribution (eV/atom) of generated MOFs, evaluated with UMA.

### G.2 Conditional generation for high $CO_2$ working capacity

We extend MOFFLOW-2 to conditional generation, targeting MOFs with high $CO_2$ working capacity. The building block generator is trained to cross-attend to a property embedding, $c = \text{Linear}(y) + \text{Fourier}(y)$, where $y$ denotes the working capacity. At inference, we sample 200 building blocks conditioned on large $y$ values and evaluate them using grand canonical Monte Carlo (GCMC) simulations [11]. As shown in Figure 9a, MOFFLOW-2 produces MOFs with higher working capacity than MOFDiff [11]. Note that MOFDiff relies on latent optimization with a fixed target of $y = 15$, while our approach conditions directly on the property.

### G.3 Energy-level evaluation using MLIP

We further evaluate the energy levels of generated MOFs using UMA [52], a state-of-the-art machine-learned interatomic potential. Specifically, we generate 10,000 structures with both MOFFLOW-2 and MOFDiff and compute the energy per atom (eV/atom) using the `uma-m-1.1` model. As shown in Figure 9b, the energy distribution of structures generated by MOFFLOW-2 aligns more closely with the training dataset than that of MOFDiff.

## H  Limitations

Although MOFFLOW-2 demonstrates strong potential for MOF structure prediction and generation, several limitations remain. Firstly, because the pipeline relies on RDKit for initial conformer generation, it is challenging to predict structures whose organic building blocks are chemically invalid or incomplete; for example, those lacking carboxylate groups, which are common in MOF decomposition schemes [37]. Second, for conditional generation in Appendix G.2, only the building block generator is conditioned on the property, whereas the structure prediction module may also depend on it. Finally, our evaluation of energy levels with machine learning interatomic potentials is less accurate than that with density functional theory (DFT).

