# OpenReview forum: "Flexible MOF Generation with Torsion-Aware Flow Matching"
_NeurIPS.cc/2025/Conference — NeurIPS 2025 poster_

### Official Review · Reviewer_eFsz · 2025-06-24

**Clarity:** 3
**Significance:** 2
**Originality:** 2
**Rating:** 4
**Confidence:** 3

**Summary:**

This paper introduces MOFFLOW-2, a flow-based generative model specifically designed for flexible Metal-Organic Framework (MOF, a family of materials)  structure prediction and design problem. The first stage leverages a SMILES-based autoregressive Transformer to generate novel chemical building blocks, while the second employs a torsion-aware flow matching model to assemble these blocks into 3D structures. MOFFLOW-2 incorporates rotations, translations, torsion angles, and lattice parameters into structure modeling. Experiments demonstrate that MOFFLOW-2 surpasses previous state-of-the-art methods in structure prediction accuracy and successfully generates novel, valid, and unique MOFs.

**Questions:**

1. The training of MOFFLOW-2 (as well as its baselines, MOFFLOW, MOFDIFF) seems to rely on a previous MOF dataset (Boyd et al.). Does the dataset itself enough for learning the distribution of MOF structure? In other words, do you think incorporating other data sources of large scale of molecule or material datasets for pretraining will help? If yes or no, why?
2. In Table 2 of structure prediction task, under smaller tolerance threshold, why the MR and RMSE seems inconsistent by contrasting DiffCSP and MOFLOW-2? Can you explain this phenomenon?
3. What is the major difference between MOFFlow-1 and -2? Which factor(s) cause the performance gain on the structure prediction task in Table 2?
4. Regarding the design problem, how was the “valid, novel and unique” defined in Figure 5, table 3 and section 6.2? I could not find the definition of these metrics.
5. By explicitly modeling the torsion angles, the modeled degree of freedom is augmented based on the model without it. Beyond that, what could be the potential “rigid” part in the MOF? In other words, which degrees of freedom are still not modeled under the setting of MOFFlow-2?

**Ethical Concerns:**

["NO or VERY MINOR ethics concerns only"]

**Final Justification:**

I have read the rebuttal/response and discussed with the authors. I think this paper is in general in good quality, albeit building upon well-established works from the domain of molecular generation and conformer prediction. The authors have responded to my question in a good manner and they basically resolved my questions, in specific, the clearness, additional experiments and some discussion regarding a broader field application. In conclusion, I lean towards borderline acceptance, where the limitation lays in its significance for much broader fields in ML.

**Limitations:**

Yes.

**Paper Formatting Concerns:**

No major formatting issues.

**Quality:**

3

**Strengths And Weaknesses:**

Strengths:
- The introduction of torsion-aware flow matching significantly advances generative modeling capabilities for MOFs by addressing flexibility explicitly.
- The paper provides extensive evaluations and comparisons with recent models, showcasing superior performance across critical metrics.
- Demonstrates the ability to generate MOFs from novel building blocks, significantly extending the design space beyond existing methods (eg. MOFFlow-1)

Weaknesses:
- Although the method is effective, the computational cost and resource requirements for larger-scale and more complex MOF generation are insufficiently discussed (for example, the training cost, inference time, etc.), which could be relevant in real-world using scenario. It could be better to include these, more or less, in the main text.
- The stability and real-world chemical feasibility of the generated (designed) MOFs, while briefly mentioned, lack comprehensive validation through computational chemistry methods (eg., DFT or ab initio calculations). That being said, it is okay to leave this for future work.
- Given MOF is a specific family of molecules, there is limited exploration of how the method scales and performs with significantly different MOF types or more chemically diverse frameworks (more general family of molecules) as an ML for science work.

---

> ### Author Rebuttal · Authors · 2025-07-30
>
> Dear reviewer eFsz,
>
> We deeply appreciate your thorough review and constructive comments on our manuscript! We also appreciate your recognition of our work, including **advancing generative modeling capabilities with torsional flexibility**, **extensive evaluations**, and **expansion of the design space with novel building blocks.** Below, we address the reviewer's concerns and questions in detail.
>
> ---
> ### W1. Insufficient discussion of computational cost and resource requirements (e.g., training cost, inference time). Include in the main text.
>
> **The training cost is detailed in Appendix A.2 and F.1, and the inference time is reported in Table 3 of the main text.** For clarity, we summarize the key information below:
>
> |   | Batch size         | GPU type    | GPU memory usage (training) | Training epochs | Total training time | Inference time (s/sample) |
> |--------|----------|------|----------|-----|------|----|
> | Building block generator (Transformer Decoder) | ~45 (<8000 tokens) | 8xA100 80GB | 50.71GB - 78.29GB           | 20              | 56m 12s             | 0.59                      |
> | Structure prediction module (Flow matching)    | ~11 (<1500 atoms)  | 8xA100 80GB | 56.67GB - 78.42GB           | 200             | 4d 11h 33m          | 1.23                      |
>
> Due to space limitations, we follow common practice and have placed details on computational resources in the Appendix. However, to address your suggestion, we will refer to this information on line 294 (Section 6) of the main text.
>
> &nbsp;
>
> ### W2. The stability of generated structures hasn't been evaluated with DFT or ab initio methods. Okay to leave for future work.
>
> We appreciate your understanding! **Nevertheless, to address your concern, we validate the energy of generated MOF structures using UMA [2], the state-of-the-art machine learning force field, which was verified to work on MOFs.** While DFT would be ideal, it is computationally infeasible for large MOF structures within the rebuttal period and our computational budget.
>
> Specifically, we generate 10,000 structures with MOFFlow-2 and MOFDiff and compute the energy per atom ($\mathrm{eV/atom}$) with the `uma-m-1.1` model. As shown in the percentile table below, MOFFlow-2 consistently produces structures with lower energy levels compared to MOFDiff. We also plan to add histograms of energy levels in the future manuscript, which could not be included in the rebuttal due to NeurIPS rules.
>
> | Model/percentile  | 20%       | 40%       | 50%       | 60%       | 80%       |
> |------|-----|----|----|----|-----|
> | $\textcolor{#bbbbbb}{\text{Reference (data)}}$ | $\textcolor{#bbbbbb}{\text{-7.06}}$     | $\textcolor{#bbbbbb}{\text{-6.86}}$     | $\textcolor{#bbbbbb}{\text{-6.78}}$    | $\textcolor{#bbbbbb}{\text{-6.68}}$    | $\textcolor{#bbbbbb}{\text{-6.48}}$   |
> | MOFDiff      | -6.60     | -6.23     | -6.00     | -5.66     | -4.54     |
> | MOFFlow-2    | **-6.75** | **-6.45** | **-6.27** | **-5.97** | **-4.56** |
>
> &nbsp;
>
> ### W3. MOF is a specific family of molecules. Limited exploration of how the method scales/performs for significantly different MOF types or more diverse frameworks as an ML for science work.
>
> **While MOF is indeed a specific class of materials, we still believe our work has huge implications.** MOFs are being investigated for many real-world problems, e.g., carbon capture, catalysis, or even drug delivery, and exhibit unique characteristics, e.g., modular nature, high-symmetry point groups, and the combination of organic and inorganic components, which require specialized algorithms.
>
> We resonate with your point and consider generalization of our work to a broader domain, e.g., including new element types or applying our model to covalent organic frameworks, as a significant research direction. We note that, in principle, our work is generalizable to such new dataset without much modification.
>
> &nbsp;
>
> ### Q1. Is the Boyd et al. dataset enough for learning the distribution of MOF structure? Will incorporating other data sources of large scale molecule/material datasets for pretraining help? Why or why not?
>
> **Since the Boyd et al. dataset [2] is a widely recognized MOF benchmark with diverse topologies and building blocks [3-5], we believe that it is a strong dataset for learning the distribution of MOFs.** However, it may not fully capture the edge cases, such as rare element types in the broader MOF space.
>
> Therefore, we believe that pretraining on large-scale molecular/material datasets can be beneficial, since the atoms in MOFs share the same fundamental interactions as in other molecules/materials. This strategy has proven effective in recent work (UMA [1]), and we expect similar benefits for our model.
>
> &nbsp;
>
> ### Q2. In Table 2, under the smaller tolerance threshold (stol=0.3), why are the MR and RMSE between DiffCSP and MOFFlow-2 inconsistent?
>
> **This inconsistency arises because RMSE is calculated only on matched structures, meaning that DiffCSP performs worse than MOFFlow-2 when considering all the generated samples.** DiffCSP has a very low match rate (\~0.08%), so its RMSE is based on a small number of trivial matches, resulting in low RMSE. In contrast, MOFFlow-2 has a much higher match rate (\~15.98%), and its RMSE is computed over a broader and more complex set of structures, resulting in a higher value.
>
> &nbsp;
>
> ### Q3. What are the major differences between MOFFlow-1 and MOFFlow-2? What causes the performance gain in the structure prediction task in Table 2?
>
> **MOFFlow-2 mainly improves over MOFFlow-1 through (1) removing the rigid body assumption and (2) adding the ability to generate MOFs from scratch.**
>
> We make a one-by-one comparison as follows:
> 1. **Rigid body assumption:** MOFFlow-1 assembles MOF structures by treating building blocks as rigid bodies and thus requires their ground-truth conformations as input. In contrast, MOFFlow-2 relaxes this assumption by explicitly modeling torsional flexibility, allowing the building blocks to adjust their conformations during assembly.
> 2. **Generative capability:** MOFFlow-1 is designed solely for structure prediction and cannot generate new MOF structures. MOFFlow-2, on the other hand, supports both generation and prediction.
>
> The performance improvement in the structure prediction task is mainly due to MOFFlow-2’s ability to model torsional flexibility, which accounts for conformational changes during the assembly.
>
> &nbsp;
>
> ### Q4. How are validity, novelty, and uniqueness (in Figure 3, Table 3, Section 6.2) defined? I could not find the definition of these metrics.**
>
> **The definitions of validity, novelty, and uniqueness are defined in lines 331-333 of the main text.** For your convenience, we restate the definition here.
> - **Validity (%):** A generated MOF is valid if it passes the MOFChecker [6] test. This includes a basic sanity check for MOFs, such as the presence of metal and organic elements, porosity, no overlapping atoms, etc.
> - **Novelty (%):** A structure is considered novel if its MOFid [7] does not appear in the training set. This essentially means that a MOF is novel if either a novel building block or a novel combination of building blocks has been used.
> - **Uniqueness (%):** Uniqueness is the proportion of distinct structures after removing duplicates from the generated set.
> - **VNU (%)** is defined as the proportion of generated structures that are simultaneously valid, unique, and novel.
>
> &nbsp;
>
> ### Q5. Which degrees of freedom have not been modeled under the setting of MOFFlow-2?
>
> **MOFFlow-2 does not model bond angles and lengths, since RDKit accurately generates them.** This is a widely accepted assumption [8, 9], which underpins many celebrated works on molecular generative models (e.g., Torsional diffusion [9], DiffDock [10], FlexDock [11]). While one could additionally model such degrees of freedom, we do not expect a meaningful improvement in accuracy despite the additional computational cost.
>
> ---
>
> We hope our experiments address your questions and concerns! Please let us know if there are any further opportunities for clarification and improving the score.
>
> ### Reference
> [1] Wood, Brandon M., et al. "UMA: A Family of Universal Models for Atoms." arXiv preprint arXiv:2506.23971 (2025).
> [2] Boyd, Peter G., et al. "Data-driven design of metal–organic frameworks for wet flue gas CO2 capture." Nature 576.7786 (2019): 253-256.
> [3] Fu, Xiang, et al. "Mofdiff: Coarse-grained diffusion for metal-organic framework design." arXiv preprint arXiv:2310.10732 (2023).
> [4] Wu, Xiaoyu, and Jianwen Jiang. "Precision-engineered metal–organic frameworks: fine-tuning reverse topological structure prediction and design." Chemical Science 15.40 (2024): 16467-16479.
> [5] Liu, Yutao, et al. "Porous framework materials for energy & environment relevant applications: A systematic review." Green Energy & Environment 9.2 (2024): 217-310.
> [6] Jablonka, K. M. (2023). mofchecker (Version 1.0.0) [Computer software]. https://doi.org/10.5281/zenodo.1234
> [7] Bucior, Benjamin J., et al. "Identification schemes for metal–organic frameworks to enable rapid search and cheminformatics analysis." Crystal Growth & Design 19.11 (2019): 6682-6697.
> [8] Axelrod, Simon, and Rafael Gomez-Bombarelli. "GEOM, energy-annotated molecular conformations for property prediction and molecular generation." Scientific Data 9.1 (2022): 185.
> [9] Jing, Bowen, et al. "Torsional diffusion for molecular conformer generation." Advances in neural information processing systems 35 (2022): 24240-24253.
> [10] Corso, Gabriele, et al. "DiffDock: Diffusion Steps, Twists, and Turns for Molecular Docking." International Conference on Learning Representations (ICLR 2023). 2023.
> [11] Corso, Gabriele, et al. "Composing unbalanced flows for flexible docking and relaxation." The Thirteenth International Conference on Learning Representations. 2025.

---

> > ### Comment · Reviewer_eFsz · 2025-08-05
> >
> > Hi authors,
> >
> > Thanks for the response which basically addressed my questions. Overall i think this paper is in good quality, albeit building upon well-established works of molecular generation and conformer prediction. Applying these techniques to the problem of MOF is interesting. Therefore, I lean towards acceptance, yet still hesitate around its significance for much broader fields (compared to diffdock, torsional diffusion, etc.). Given these factors, I would keep my current rating.

---

> ### Author Response · Authors · 2025-08-07
>
> We’re glad to hear that our response addressed your questions and concerns! Thank you also for your kind words and support for our work. We truly appreciate your constructive feedback throughout the review process, which has helped us improve the overall quality of our work.

---

### Official Review · Reviewer_Figq · 2025-07-02

**Clarity:** 3
**Significance:** 2
**Originality:** 2
**Rating:** 4
**Confidence:** 3

**Summary:**

The paper introduced MOFFLOW-2, a two-stage generative framework for metal–organic frameworks that (1) generates metal clusters and organic linkers via an autoregressive SMILES model, and (2) assembles them into full 3D crystal structures by explicitly modelling translations, rotations, lattice, and torsional degrees of freedom using Riemannian flow matching.  The method has been evaluated on two tasks: structure prediction and generation.

**Questions:**

Regarding the initial 3D structure of linkers, did you investigate the model's sensitivity to the quality of the RDKit-generated conformer? Can you comment more on that?

**Ethical Concerns:**

["NO or VERY MINOR ethics concerns only"]

**Final Justification:**

I'm supportive in general but I still see some limitations such as the baselines used for comparison.

**Limitations:**

Yes.

**Paper Formatting Concerns:**

No.

**Quality:**

3

**Strengths And Weaknesses:**

**Strengths**

- Two-stage framework: I think the idea of two-stage decomposition, first smiles generation, and then capturing 3D structure via flow matching is an effective solution for this task.
- Empirical Evaluation: The authors evaluated their method on two tasks and the method seems to has better RMSE in structure prediction compared to previous baselines.

**Weaknesses**

- I think the evaluation is still limited, and the authors compared to classical methods and a few baselines like MOFDiff.
- The method still consider the use of bioinformatics tools like RDKit, as the second stage relies on an initial 3D structure for the organic linkers generated by RDKit. Also, while the model learns to correct torsion angles, it still assumes the initial bond lengths and angles are accurate.

---

> ### Author Rebuttal · Authors · 2025-07-30
>
> Dear reviewer Figq,
>
> We deeply appreciate your thorough review and constructive comments on our manuscript! We also appreciate your recognition of our work, including **the effectiveness of our two-stage approach** and  **improved empirical results**. Below, we address the reviewer's concerns and questions in detail.
>
> ---
> ### W1. I think the evaluation is still limited, and the authors compared to classical methods and a few baselines like MOFDiff.
>
> **We consider our set of baselines to be comprehensive, given how the MOF generative model is an under-explored research area.** We note that MOFFlow-1 and MOFDiff are among the first deep learning models that can generate MOFs from scratch. MOFDiff itself had no learning-based baselines for similar reasons.
>
> We also note that classical methods like random search (RS) and evolutionary algorithms (EA) are strong and widely used baselines in the crystal structure prediction (CSP) field, making them meaningful baselines for comparison.
>
> If you have any suggestions for the baselines, we would be happy to incorporate them.
>
> &nbsp;
>
> ### W2. Method relies on a bioinformatics tool like RDKit for initial structure generation, and assumes that the initial bond lengths and angles are accurate.
>
> **We do not think this is a weakness of our method, as the bond lengths and angles generated by cheminformatics toolkits are widely accepted to be accurate and accessible [1, 2].** Our algorithm re-initializes the remaining structure (torsion) and does not depend on highly variable information. Also, the reliance on RDKit for estimating the bond lengths and angles underpins many celebrated works on molecular generative models (e.g., Torsional diffusion [2], DiffDock [3], FlexDock [4]).
>
> &nbsp;
>
> ### Q1. Did you investigate the model's sensitivity to the quality of RDKit-generated conformers?
>
> **Yes; MOFFlow-2 is robust to the quality of RDKit-generated conformers, since it relies only on the bond lengths and angles, which RDKit reliably estimates.** While RDKit may be inaccurate for torsion angles, our model is insensitive to this since it randomly reinitializes them at $t=0$.
>
> To empirically confirm this, we evaluate our model on four sets of initial conformers randomly generated by RDKit, given the same building blocks. As shown below, the performance remains consistent, with negligible variation in both validity and VNU metrics:
>
> |            | Valid (%)        | VNU (%)          |
> |------------|------------|------------------|
> | Trial 1    | 37.6             | 31.0             |
> | Trial 2    | 37.4             | 30.6             |
> | Trial 3    | 37.3             | 30.5             |
> | Trial 4    | 37.5             | 30.8             |
> | **Mean ± Std** | **37.45 ± 0.11** | **30.72 ± 0.19** |
>
> ---
>
> We hope our experiments address your questions and concerns! Please let us know if there are any further opportunities for clarification and improving the score.
>
> ### Reference
> [1] Axelrod, Simon, and Rafael Gomez-Bombarelli. "GEOM, energy-annotated molecular conformations for property prediction and molecular generation." Scientific Data 9.1 (2022): 185.
> [2] Jing, Bowen, et al. "Torsional diffusion for molecular conformer generation." Advances in neural information processing systems 35 (2022): 24240-24253.
> [3] Corso, Gabriele, et al. "DiffDock: Diffusion Steps, Twists, and Turns for Molecular Docking." International Conference on Learning Representations (ICLR 2023). 2023.
> [4] Corso, Gabriele, et al. "Composing unbalanced flows for flexible docking and relaxation." The Thirteenth International Conference on Learning Representations. 2025.

---

> > ### Comment · Reviewer_Figq · 2025-08-08
> >
> > I thank the authors for their response. I'll maintain my score as I'm already supportive.

---

> > > ### Author Response · Authors · 2025-08-09
> > >
> > > Thank you for your kind support of our work. We truly appreciate your constructive comments throughout the review process, which have helped us improve the overall quality of our work.

---

### Official Review · Reviewer_EcT6 · 2025-07-03

**Clarity:** 2
**Significance:** 2
**Originality:** 2
**Rating:** 4
**Confidence:** 3

**Summary:**

This paper introduces MOFFLOW-2, a two-stage generative framework for metal–organic framework (MOF) design. The first stage leverages an autoregressive SMILES-based language model to generate novel metal and organic building blocks. The second stage employs a flow matching model to predict the 3D MOF structure by modeling translations, rotations, torsions, and lattice parameters. This approach addresses key limitations of prior work that relied on fixed, rigid building blocks.

**Questions:**

- How sensitive is the performance of MOFFLOW-2 to the initial RDKit geometries used for organic linkers?
- How well does the model generalize to MOF chemistries that are entirely out-of-distribution from training?

**Ethical Concerns:**

["NO or VERY MINOR ethics concerns only"]

**Limitations:**

- The physical plausibility of the generated MOFs is not validated using quantum or force-field level simulations.
- The framework does not show results in property-conditioned or goal-directed generation, which may limit its applicability in practical design tasks.

**Quality:**

2

**Strengths And Weaknesses:**

**Strengths:**

- Clear motivation and reasonable contribution: The paper addresses critical limitations of prior MOF generative models by enabling the generation of novel building blocks and incorporating torsional flexibility.
- Substantial empirical improvements: On structure prediction, MOFFLOW-2 outperforms all tested baselines under strict matching criteria. On generative design, it shows higher validity, novelty, uniqueness, and property diversity.
- Good writing and clarity: The paper is well-structured, with clear diagrams, descriptions, and motivation throughout.

**Weaknesses:**
- Lack of physical validation: The generated structures are not assessed using quantum mechanical (e.g., DFT) or molecular dynamics simulations, which limits confidence in their physical stability.
- Limited task scope: The framework is not demonstrated in property-driven generation scenarios, which would be critical for targeted MOF design.

---

> ### Author Rebuttal · Authors · 2025-07-30
>
> Dear reviewer EcT6,
>
> We deeply appreciate your thorough review and constructive comments on our manuscript! We also appreciate your recognition of our work, including **clear motivation and reasonable contribution**, **substantial empirical improvements**, and **good writing clarity.** Below, we address the reviewer's concerns and questions in detail.
>
> ---
> ### W1. Lack of physical validation.
>
> **To address your concern, we validate the energy of generated MOF structures using UMA [1], the state-of-the-art machine learning force field, which was verified to work on MOFs.** While DFT would be ideal, it is computationally infeasible for large MOF structures within the rebuttal period and our computational budget.
>
> Specifically, we generate 10,000 structures with MOFFlow-2 and MOFDiff and compute energy per atom ($\mathrm{eV/atom}$) with the `uma-m-1.1` model. As shown in the percentile table below, MOFFlow-2 consistently produces structures with lower energy levels compared to MOFDiff. We also plan to add histograms of energy levels in the future manuscript, which could not be included in the rebuttal due to NeurIPS rules.
>
> | Model/percentile  | 20%       | 40%       | 50%       | 60%       | 80%       |
> |--------------|-----------|-----------|-----------|-----------|-----------|
> | $\textcolor{#bbbbbb}{\text{Reference (data)}}$ | $\textcolor{#bbbbbb}{\text{-7.06}}$     | $\textcolor{#bbbbbb}{\text{-6.86}}$     | $\textcolor{#bbbbbb}{\text{-6.78}}$    | $\textcolor{#bbbbbb}{\text{-6.68}}$    | $\textcolor{#bbbbbb}{\text{-6.48}}$   |
> | MOFDiff      | -6.60     | -6.23     | -6.00     | -5.66     | -4.54     |
> | MOFFlow-2    | **-6.75** | **-6.45** | **-6.27** | **-5.97** | **-4.56** |
>
> &nbsp;
>
> ### W2. The framework does not demonstrate property-driven generation scenarios.
>
> We appreciate the reviewer’s insightful comment. **Our method can be easily extended to property-driven generation, and here we demonstrate this by generating MOFs to maximize CO2 working capacity.** To this end, we (1) train MOFFlow-2 conditioned on CO2 working capacity and (2) generate MOFs conditioned on sufficiently high values of this property.
>
> To be specific, we re-train the building block generator to cross-attend to working capacity $y$ featurized as $c=\text{Linear}(y) + \text{Fourier}(y)$. During inference, we sample 200 building blocks conditioned on $y=4$ and compute the CO2 working capacity by running a grand canonical Monte Carlo (GCMC) simulation [2]. As shown in the table below, MOFFlow-2 successfully generates MOFs with high working capacity and outperforms MOFDiff. Note that MOFDiff applies latent optimization with a target of $y=15$.
>
> |                              | Working capacity vacuum swing (mol/kg) |
> |------------------------------|----------------------------------------|
> | Training data                | 0.4242                                 |
> | MOFDiff (w/ optimization)    | 1.1798                                 |
> | MOFFlow-2 (w/ conditioning)  | **2.1694**                             |
>
> We also plan to update our code and add working capacity histograms in the future manuscript, which could not be included in the rebuttal due to NeurIPS rules.
>
> &nbsp;
>
> ### Q1. How sensitive is the performance of MOFFlow-2 to the initial RDKit geometries used for organic linkers?
>
> **MOFFlow-2 performs robustly regardless of initial RDKit geometries, since it relies only on the bond lengths and angles, which RDKit reliably estimates.** While RDKit may be inaccurate for torsion angles, our model is insensitive to this since it randomly reinitializes them at $t=0$.
>
> To empirically confirm this, we evaluate our model on four sets of initial conformers randomly generated by RDKit, given the same building blocks. As shown below, the performance remains consistent, with negligible variation in both validity and VNU metrics:
>
> |            | Valid (%)        | VNU (%)          |
> |------------|------------------|------------------|
> | Trial 1    | 37.6             | 31.0             |
> | Trial 2    | 37.4             | 30.6             |
> | Trial 3    | 37.3             | 30.5             |
> | Trial 4    | 37.5             | 30.8             |
> | **Mean ± Std** | **37.45 ± 0.11** | **30.72 ± 0.19** |
>
> &nbsp;
>
> ### Q2. How well does the model generalize to MOF chemistries that are entirely out-of-distribution (OOD) from training?
>
> **Similar to any other machine learning algorithm, our method will struggle to generalize to data that is entirely out-of-distribution (OOD).** For example, MOFFlow-2 is not capable of generalizing to unseen element types. Also, its performance for MOFs with complex topologies and many building blocks may be lower than those common in the dataset. However, this can be alleviated by training our model on a more diverse set of MOF/crystal structures (e.g., datasets from [3] or the Materials Project [4]).
>
> ---
>
> We hope our experiments address your questions and concerns! Please let us know if there are any further opportunities for clarification and improving the score.
>
> ### Reference
> [1] Wood, Brandon M., et al. "UMA: A Family of Universal Models for Atoms." arXiv preprint arXiv:2506.23971 (2025).
> [2] Fu, Xiang, et al. "Mofdiff: Coarse-grained diffusion for metal-organic framework design." arXiv preprint arXiv:2310.10732 (2023).
> [3] Lee, Sangwon, et al. "Computational screening of trillions of metal–organic frameworks for high-performance methane storage." ACS Applied Materials & Interfaces 13.20 (2021): 23647-23654.
> [4] Jain, Anubhav, et al. "Commentary: The Materials Project: A materials genome approach to accelerating materials innovation." APL materials 1.1 (2013).

---

> ### Comment · Area_Chair_SNjf · 2025-08-08
> **Response to rebuttal needed**
>
> @Reviewer EcT6: Please respond to authors rebuttal and update rating accordingly.

---

### Official Review · Reviewer_qUfU · 2025-07-03

**Clarity:** 3
**Significance:** 3
**Originality:** 3
**Rating:** 4
**Confidence:** 2

**Summary:**

This paper introduces MOFFLOW-2, a two-stage generative framework designed for the prediction of metal-organic framework (MOF) structures and the creation of novel MOFs. The framework comprises two main components: (1) an autoregressive SMILES-based model for generating metal and organic building blocks, and (2) a flow-matching model for assembling these blocks into valid 3D structures by predicting translations, rotations, torsional angles, and lattice parameters. The approach addresses limitations of previous models by generating novel building blocks, predicting flexible 3D conformations, and not relying on a fixed set of building blocks or rigid conformations. Experimental results show that MOFFLOW-2 outperforms previous methods like MOFDiff and MOFFLOW in both structure prediction accuracy and the generation of novel, valid, and unique MOFs.

**Questions:**

See the weakeness.

**Ethical Concerns:**

["NO or VERY MINOR ethics concerns only"]

**Final Justification:**

I thank the authors for the additional experiments. I will maintain my score, which inclines towards acceptance.

**Limitations:**

Yes, the authors have included a limitation section in their paper.

**Paper Formatting Concerns:**

No.

**Quality:**

3

**Strengths And Weaknesses:**

Strengths:
+ **Novel Approach**: The introduction of a two-stage framework that combines SMILES-based building block generation with flexible 3D structure prediction is innovative and allows for greater chemical diversity and structural flexibility.

+ **Strong Experimental Results**: The model outperforms baseline methods in both structure prediction and MOF generation tasks, showing improved validity, novelty, and uniqueness in generated structures. This is particularly important for discovering new MOFs and expanding chemical space.

+ **Clear Methodology and Comprehensive Architecture**: The authors present a clear and detailed explanation of their methodology, including model architecture, preprocessing steps, and the flow-matching process for predicting rotational, translational, and torsional transformations.


Weaknesses:


+ **Insufficient Ablation Studies**: A key concern is the absence of a clear breakdown of the contributions from individual model components (e.g., the SMILES generator and flow-matching) to overall performance. Since the generation process is factorized into two stages, it is crucial to independently analyze the contribution of each stage and identify potential bottlenecks. Is it possible to provide experiments that offer insight into this aspect?

---

> ### Author Rebuttal · Authors · 2025-07-30
>
> Dear reviewer qUfU,
>
> We deeply appreciate your thorough review and constructive comments on our manuscript! We also appreciate your recognition of our work, including **the novelty of our approach**, **strong experimental results**, and **clear methodology/architecture.** Below, we address the reviewer's concern in detail.
>
>  ---
>
> ### W1. Insufficient ablation studies. Analyze the contribution of each component in the two-stage pipeline (e.g., the SMILES generator and flow matching) to the overall performance.
>
> Thank you for your insightful comment. **While our comparison of MOFFlow-2 and MOFFlow-1 in Table 2 ablates the effect of torsion flexibility for structure prediction, we provide more studies to resolve your concern.** Our new results confirm that both (1) SMILES generation quality and (2) torsional flexibility modeling are important for MOF generation.
>
> **1. Varying SMILES generator quality:** We first ablate the effect of the SMILES generator on the MOF generation quality. To this end, we train the SMILES generator with a varying number of steps and evaluate its quality with SMILES-VNU* (left column). Each is then paired with the same flow matching module, and the full pipeline is evaluated using VNU (right column). Results show that improvements in the SMILES generator directly lead to higher overall MOF quality.
>
> **SMILES-VNU is the percentage of generated MOF sequences that are valid (RDKit sanity check), novel (not in training set), and unique (non-duplicated).*
>
> | SMILES quality (SMILES-VNU%) | Overall MOF quality (VNU %)|
> |--------------------------|---------------|
> | 45.9                     | 17.90         |
> | 58.6                     | 19.11         |
> | 69.0                     | 22.32         |
> | 84.6                     | 31.26         |
> | **85.3**                | **31.35**     |
>
> **2. Excluding torsional flexibility:** Next, we ablate the effect of torsional flexibility on the MOF generation quality. Replacing MOFFlow-2 with MOFFlow-1 (no torsion modeling) in the assembly stage clearly degrades the performance in MOF generation.
>
> |           | Valid (%) | VNU (%)   |
> |-----------|-----------|-----------|
> | SMILES gen + FM (w/o torsion) | 29.76     | 23.81     |
> | SMILES gen + FM (w/ torsion, ours) | **38.84** | **31.35** |
>
> ---
>
> We hope our experiments address your concerns! Please let us know if there are any further opportunities for clarification and improving the score.

---

> > ### Comment · Reviewer_qUfU · 2025-08-06
> >
> > Thank you for the additional experiments, which are valuable supplements to the study. I will maintain my score, which inclines towards acceptance.

---

> ### Author Response · Authors · 2025-08-07
>
> We’re happy to hear that the ablation studies addressed your concerns. Thank you as well for your kind support of our work. We truly appreciate your constructive comments throughout the review process, which have helped us improve the overall quality of our work.

---

### Note · Authors · 2025-08-13

Dear Area Chair and Reviewers,

We sincerely thank you for your time, thoughtful feedback, and guidance throughout the review process. We are encouraged by how the reviewers generally recommend acceptance of our work (score: 4/4/4/4) and acknowledges that our work introduces a novel approach (qUfU) with clear motivation (EcT6) that advances generative modeling for MOFs (eFsz), supported by strong empirical results (qUfU, EcT6, Figq, eFsz).

We are happy to hear that our rebuttal resolved the reviewers' concerns. Nevertheless, to assist in your final evaluation, we summarize below the key discussion points and improvements made during the rebuttal.

1. **Lack of physical validation:** We showed that our model outperforms MOFDiff using a machine learning force field (UMA).
2. **Lack of property-conditioned generation:** We trained MOFFlow-2 conditioned on CO2 working capacity. The new model outperforms MOFDiff for generating MOFs with high CO2 working capacity.

All rebuttal clarifications and new experimental results will be incorporated into the final manuscript. We believe the rebuttal process has substantially strengthened our work.

Thank you once again for your constructive feedback and guidance.

Sincerely,
Authors.

---

### Decision · Program_Chairs · 2025-09-17

**Decision:**

Accept (poster)

**Comment:**

The paper presents a new 2-stage method for generating MOFs. All reviewers rated it as borderline accept, recognizing the technical novelty and strong empirical results while raising concerns about physical validation and property conditioning that were effectively addressed in a comprehensive rebuttal. The work makes a solid technical contribution to an important materials science problem, and I recommend acceptance.